# Age-dependent Pdgfrβ signaling drives adipocyte progenitor dysfunction to alter the beige adipogenic niche in male mice

Abigail M. Benvie[1], Derek Lee [ORCID][1], Benjamin M. Steiner[1], Siwen Xue[1], Yuwei Jiang[2] & Daniel C. Berry [ORCID][1] ✉

Perivascular adipocyte progenitor cells (APCs) can generate cold temperature-induced thermogenic beige adipocytes within white adipose tissue (WAT), an effect that could counteract excess fat mass and metabolic pathologies. Yet, the ability to generate beige adipocytes declines with age, creating a key challenge for their therapeutic potential. Here we show that ageing beige APCs overexpress platelet derived growth factor receptor beta (*Pdgfrβ*) to prevent beige adipogenesis. We show that genetically deleting *Pdgfrβ*, in adult male mice, restores beige adipocyte generation whereas activating *Pdgfrβ* in juvenile mice blocks beige fat formation. Mechanistically, we find that Stat1 phosphorylation mediates Pdgfrβ beige APC signaling to suppress *IL-33* induction, which dampens immunological genes such as *IL-13* and *IL-5*. Moreover, pharmacologically targeting Pdgfrβ signaling restores beige adipocyte development by rejuvenating the immunological niche. Thus, targeting Pdgfrβ signaling could be a strategy to restore WAT immune cell function to stimulate beige fat in adult mammals.

Mammalian ageing is accompanied by an increased occurrence in obesity and metabolic dysfunction, shortening longevity and hindering the quality of life[1]. Central to the obesity epidemic is the uncontrolled expansion and accumulation of white adipose tissue (WAT), which can foster type 2 diabetes, cardiometabolic diseases, and chronic inflammation[2–6]. Hence, finding inherent mechanisms to burn excess energy to avert age-associated WAT growth could be exploited to attenuate these pathologies and to rejuvenate systemic metabolism in ageing individuals. For example, in response to cold temperature exposure, WAT resident perivascular smooth muscle-like adipocyte progenitor cells (APCs) undergo beige adipocyte development[7,8]. Once generated, beige adipocytes futilely burn circulating glucose and free fatty acids to generate heat, an effect that could counteract WAT expansion and improve metabolism[9–11]. Consistent with this notion, augmenting the number of beige adipocytes or their activity promotes a healthy metabolic profile (i.e., reducing adiposity and lowering blood sugar and fats) in lean mammals[12]. Conversely, lowering the amount or

activity of thermogenic fat can predispose rodents, and potentially humans, to metabolic dysregulation in response to a positive energy balance[13].

In addition to beige adipocyte recruitment, cold temperatures also facilitate WAT remodeling, which can induce changes in adipocyte size, vascularization, and cellular composition[14]. In particular, cold exposure can invoke alterations in WAT immune cell composition, which have been suggested to coordinate beige fat development[15–18]. For instance, changes in the abundance of alternatively activated macrophages and group 2 innate lymphoid cells (ILC2s) have been shown to potentially regulate catecholamine levels and APC proliferation and differentiation, respectively, driving beige fat formation and activity[15–18]. Yet, overriding the prospective health benefits of thermogenic tissue is the inability to generate cold-induced beige adipocytes with age[19–21]. The beiging process, markedly, begins to fail in humans as early as their mid-30's, creating a significant clinical challenge for the therapeutic promise of beige adipose, especially

[1]Division of Nutritional Sciences, Cornell University, Ithaca, NY 14853, USA. [2]Department of Physiology and Biophysics, University of Illinois at Chicago, Chicago, IL 60612, USA. ✉e-mail: dcb37@cornell.edu

within the ageing obese population[20]. Moreover, the mechanisms driving age-dependent beige adipogenic failure is not entirely understood[22].

The functional decline in tissue maintenance and function is often due to stem cell or progenitor ageing or senescence[23]. Generally, cellular senescence is characterized by a state of cellular arrest and the acquisition of a secreted pro-inflammatory program that targets neighboring cell types, thereby disrupting tissue homeostasis[24,25]. Thus, the age-related decline in stem cell behavior and regenerative potential due to cellular senescence dampens the overall ability for cell and tissue replacement in older organisms. Comparably, the loss of beige adipogenic potential appears to be partly due to the acquisition of a cellular senescence-like-phenotype of beige APCs, rendering them unresponsive to cold temperatures and non-adipogenic[19]. In several senescence-acquiring tissues, the functional decline in stem cell activity leads to changes in the tissue microenvironment or niche. Alterations to the cellular composition and/or changes in niche signaling can influence organ homeostasis, ultimately, leading to decline in its function[26,27]. Yet, our understanding of the physiological and molecular consequences of ageing beige APCs on the WAT microenvironment have yet to be realized[28].

In this work, we probed the notion that ageing beige APCs modify the WAT niche to disrupt cold-induced beige fat formation in ageing mammals. Towards this end, we find an age-dependent upregulation in platelet-derived growth factor receptor beta (*Pdgfrβ*) expression and signaling within beige APCs. The data suggest that Pdgfrβ signaling is a negative rheostat for beige fat development in adult mammals. This elevation in Pdgfrβ signaling via Stat1 appears independent of cellular senescence. Alternatively, Pdgfrβ-Stat1 signaling downregulates *IL-33* expression, impairing type 2 cytokine signaling. Genetically or pharmacologically manipulating the Pdgfrβ-Stat1 signaling restores beige adipogenic potential and cold-induced immune cell activation in aged mice.

## Results

### *Pdgfrβ* expression is elevated within aged beige APCs

Mammalian ageing has been shown to impair cold temperature-induced beige fat development[19,21,22,27,29–31]. Indeed, we observed a marked reduction in the development of cold temperature (6.5 °C) induced Ucp1+ beige adipocytes within dorsolumbar inguinal WAT[32] (iWAT) sections from 6-, 12-, and 24-month-old mice compared to 2-month-old mice (Fig. 1a, b and Supplementary Fig. 1a, b). To gain molecular insight into how ageing might impact beige adipogenic potential, we performed RNA sequencing (RNA-seq) on the stromal vascular fraction (SVF) from iWAT depots from 2-, and 12-month-old mice maintained at room temperature (RT). Notably, we observed transcriptional changes of over 3400 genes between 2- and 12-month-old SVF (Fig. 1c, Supplementary Fig. 1c, and Supplementary Data 1 and 2). Consistent with our previous study[19], we observed age-dependent changes in *p38/Mapk* and *p53* signaling pathways, suggesting cellular ageing of WAT as a potential regulator of beige adipogenic failure (Supplementary Fig. 1c–e and Supplementary Data 1 and 2). In addition, we found an age-associated upregulation *Pdgfrβ*, a membrane-bound receptor tyrosine kinase[33,34], which marks APC-mural cells (Supplementary Fig. 1d, e)[7,8,35,36]. Interestingly, *Pdgfrβ* appears to have a regulatory role in WAT development and homeostasis but the functional requirement of *Pdgfrβ* to control beige fat biogenesis and its link to ageing is unclear[37–39].

We validated the RNA-seq data by performing Pdgfrβ immunoblotting on iWAT SVF isolated from 2- and 12-month-old mice and indeed, Pdgfrβ protein was upregulated in an age-associated manner (Supplementary Fig. 1f, g). Because Pdgfrβ marks APCs, we investigated if age-related changes in *Pdgfrβ* expression could be detected within the α-smooth muscle actin (Sma/Acta2) marked beige APC lineage compared to the surrounding stroma (Supplementary Fig. 1h).

To do so, we employed the *Sma-Cre^ERT2; R26^-mTmG* (*Sma*-Control) beige APC lineage marking and tracking genetic tool[8,40]. To induce recombination, we administered one dose of tamoxifen (TMX) for 2 consecutive days to 2-, 6-, and 12-month-old *Sma*-Control mice (Supplementary Fig. 1i). Notably, after TMX-induction, ~100% of sort-purified mGFP cells were Sma+ and ~75% of the total Sma-antibody+ cellular pool was mGFP+, suggesting robust recombination efficiency[8] (Supplementary Fig. 1j–m). In agreement with the RNA-seq data, we found that *Pdgfrβ* expression displayed an age-dependent (2, 6, and 12 months of age) linear upregulation within beige APCs, analogous to *p16^Ink4a* (Fig. 1d). Further, flow cytometric analysis of Sma+ cells, verified an age-dependent upregulation of Pdgfrβ protein (Supplementary Fig. 1n).

To examine if these ageing observations extended beyond rodents, we assessed human SV cells from non-obese BMI matched vicenarian (23.5 ± 3.87 years; BMI: 23.43 ± 4.32) and quadragenarian (49.33 ± 6.02 years; BMI: 24.67 ± 2.6) patients[19]. We found that human *Pdgfrβ* mRNA expression was increased in quadragenarian SVF samples compared to vicenarians (Fig. 1e). Together, these data suggest that *Pdgfrβ* expression is elevated with iWAT ageing which may function to regulate beige adipocyte biogenesis.

### *Pdgfrβ* mediates age-dependent beige adipogenic failure

Age-related changes in *Pdgfrβ* mRNA expression led us to examine if this signaling pathway was involved in the functional decline of beige fat. To test if *Pdgfrβ* controlled beige adipogenic potential in vivo, we combined the *Pdgfrβ^fl/fl* conditional mouse model with the *Sma*-Control mouse model (*Sma-Cre^ERT2; R26^-RFP or mTmG)[41,8,40]* to create *Sma*-Control (Control) and *Sma-Pdgfrβ*-KO (*Pdgfrβ*-KO) mice[37] (Fig. 2a). To induce *Pdgfrβ* deletion, we administered one dose of TMX (50 mg/kg) for 2 consecutive days to 2-, 6-, and 12-month-old control and *Pdgfrβ*-KO male mice. Subsequently, mice were randomized to RT or cold exposure for 7 days (Fig. 2b)[8]. Notably, *Pdgfrβ* mRNA expression was reduced in FACS-isolated mutant Sma+ cells (Supplementary Fig. 2a). Control and mutant mice maintained at RT, regardless of age, did not reveal changes in beige fat development or thermogenic genes within the dorsolumbar iWAT region (Supplementary Fig. 2b–e). In response to cold temperature exposure, 2-month-old control and mutant mice appeared to generate equivalent responses in body temperature defense, adiposity, histological morphology, Ucp1 immunostaining presence, and thermogenic gene induction (Fig. 2c, d and Supplementary Fig. 3a–c). In stark contrast, cold-exposed *Pdgfrβ*-KO mutant mice at 6 and 12 months of age were better able to defend their body temperature and had lower total adiposity (combined subcutaneous and visceral depots) than age-matched cold-exposed control littermates (Supplementary Fig. 3d–i). Furthermore, histological analysis revealed an abundance of Ucp1+ beige adipocytes within cold exposed 6- and 12-month-old *Pdgfrβ*-KO mutant dorsolumbar iWAT sections (Fig. 2c, d and Supplementary Fig. 3j). Consistent with beige adipocyte formation, directed qPCR analysis showed a robust induction of beige and thermogenic genes in 6- and 12-month-old *Pdgfrβ*-KO iWAT compared to age-matched controls (Supplementary Fig. 3f, i). In agreement with our previous fate mapping and physiological studies[8,42], brown adipose tissue (BAT) appeared to be unaffected by the *Sma*-driven *Pdgfrβ* deletion as assessed by weight, morphology, and *Ucp1* gene expression (Supplementary Fig. 3k–m).

### Activating Pdgfrβ causes premature beige adipogenic failure

As a next step, we examined if Pdgfrβ signaling was sufficient to prematurely block beige adipogenesis in 2-month-old mice. Towards this end, we activated Pdgfrβ by administering recombinant Pdgf-BB (25 ng/mouse)[43] to 2-month-old RT housed male control mice for 5 consecutive days (Supplementary Fig. 4a, b). Subsequently, mice were cold exposed for 3 days—to capture initial beiging events in young mice—without ligand (Fig. 3a). In response to acute cold exposure, we

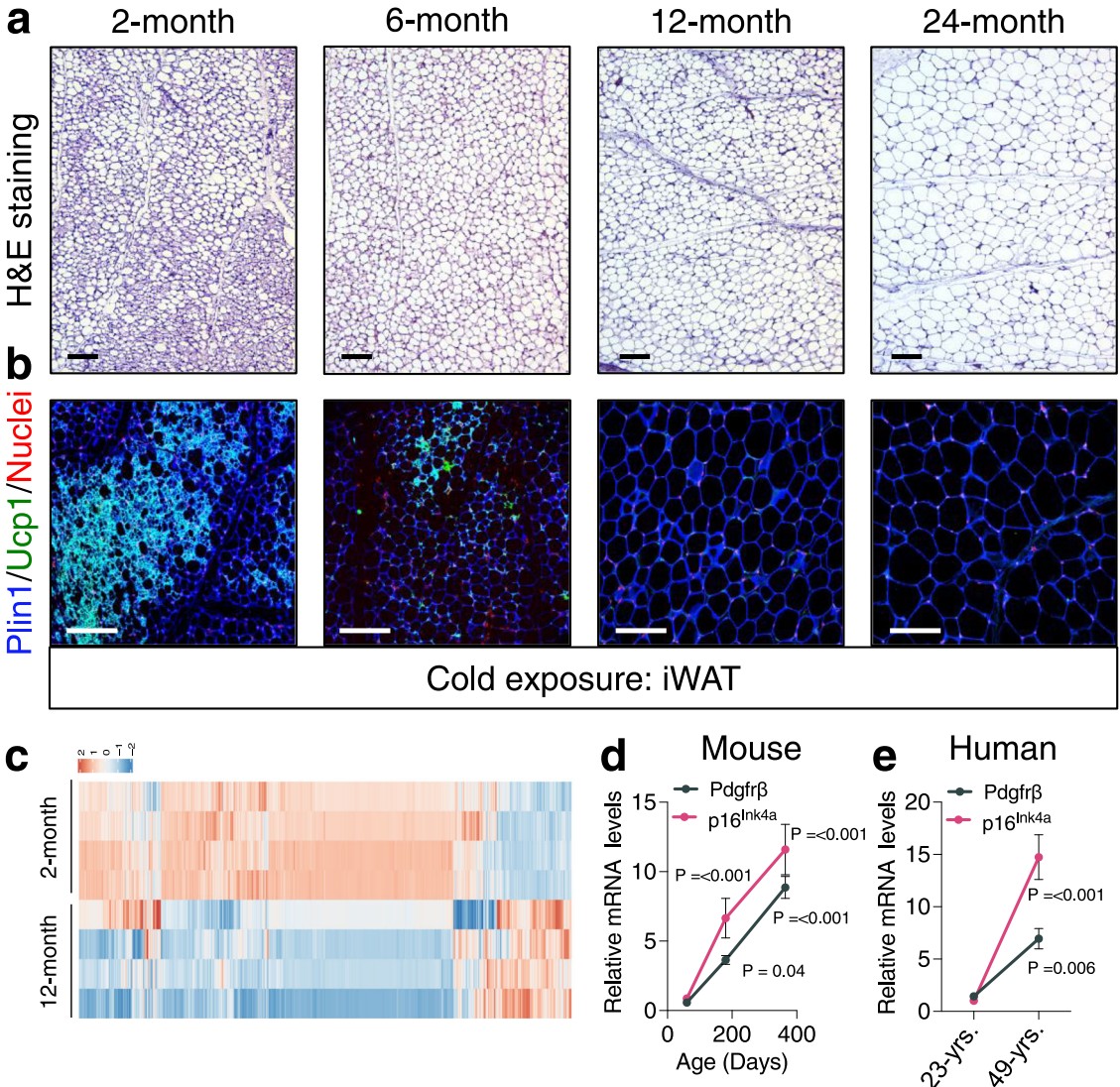

**Fig. 1 | Ageing is associated with diminished beige fat development and Pdgfrβ upregulation. a, b** Representative H&E staining (**a**) and perilipin (Plin1; blue) and Ucp1 (green) immunostaining (**b**) of dorsolumbar iWAT sections from cold exposed (6.5 °C for 7 days) 2-, 6-, 12-, and 24-month-old C57BL6/J-129SV male mice (×10; ×20 magnification, scale bars 100 μm). **c** Heatmap of gene expression profiles comparing iWAT SVF from 2- and 12-month-old male mice maintained at RT.

**d, e** mRNA levels of Pdgfrβ and p16[Ink4a] gene expression in FACS isolated iWAT Sma + beige APCs at denoted ages (**d**) or from vicenarian and quadragenarian human WAT SVF (**e**) (n = 4 mice or humans/group). Data are presented as mean values ± SEM. Data were analyzed by two-tailed Student's t-test. Source data are provided within the Source Data file. The full list of genes and normalized counts for the gene expression analysis can be found in Supplementary Data 1 and 2.

observed that Pdgf-BB treatment reduced beige adipocyte generation as assessed by H&E staining, Ucp1 immunostaining, and multilocular adipocyte quantification (Fig. 3b–d). Moreover, Pdgf-BB treatment prevented the induction of beige genetic markers (Fig. 3e). These effects of Pdgf-BB treatment on beige fat appearance seemed to be independent of changes in BAT weight, morphology, and gene expression (Supplementary Fig. 4c–e).

To genetically assess if activating Pdgfrβ signaling within beige APCs was sufficient to disrupt cold-induced beige adipocyte formation, we generated a mouse model (*Sma-CreERT2*; *R26-mTmG*, *PdgfrβD849V*) with *Sma*-dependent inducible expression of a constitutively active *Pdgfrβ* allele (*PdgfrβD849V*)[37,44] (Supplementary Fig. 4f). Notably, the *PdgfrβD849V* knockin mouse model harbors a point mutation within the kinase domain conferring constitutive activation but only in the presence of active Cre[44]. Indeed, after TMX induction and recombination, we observed elevated basal Pdgfrβ phosphorylation within the SVF compartment from *PdgfrβD849V* iWAT (Fig. 3f and Supplementary Fig. 4g, h). In response to cold temperatures, constitutive activation of Pdgfrβ hindered the development of Ucp1+ beige adipocytes within

iWAT depots. (Fig. 3g–i). In agreement, gene expression profiling of iWAT depots from *PdgfrβD849V* mice showed a diminished induction of thermogenic genes (Fig. 3j). Consistent, BAT weight, morphology, and gene expression appeared unchanged between control and constitutively active Pdgfrβ mutant mice (Supplementary Fig. 4i–k). Given that the *Sma* promoter is expressed within all vascular smooth muscle cells throughout the body, it is reasonable that Pdgfrβ functions in other cell types, contributing to changes in beige fat development[33]. Nevertheless, the observed reciprocal effects of Pdgfrβ gain- and loss-of-function strongly suggest that Pdgfrβ action within beige APCs can modulate beige fat biogenesis in an age-related manner.

**Pdgfrβ does not mediate beige APC cellular senescence**

The age-dependent decline in cold-induced beige adipogenic potential is, in part, due to a cellular senescence-like phenotype of beige APCs. A molecular mediator of beige APC cellular aging is p38 phosphorylation[19]. We hypothesized that ablating Pdgfrβ signaling might reduce the phosphorylation status of p38/Mapk within 12-month-old beige APCs to restore adipogenic potential[19,34]. Consistent

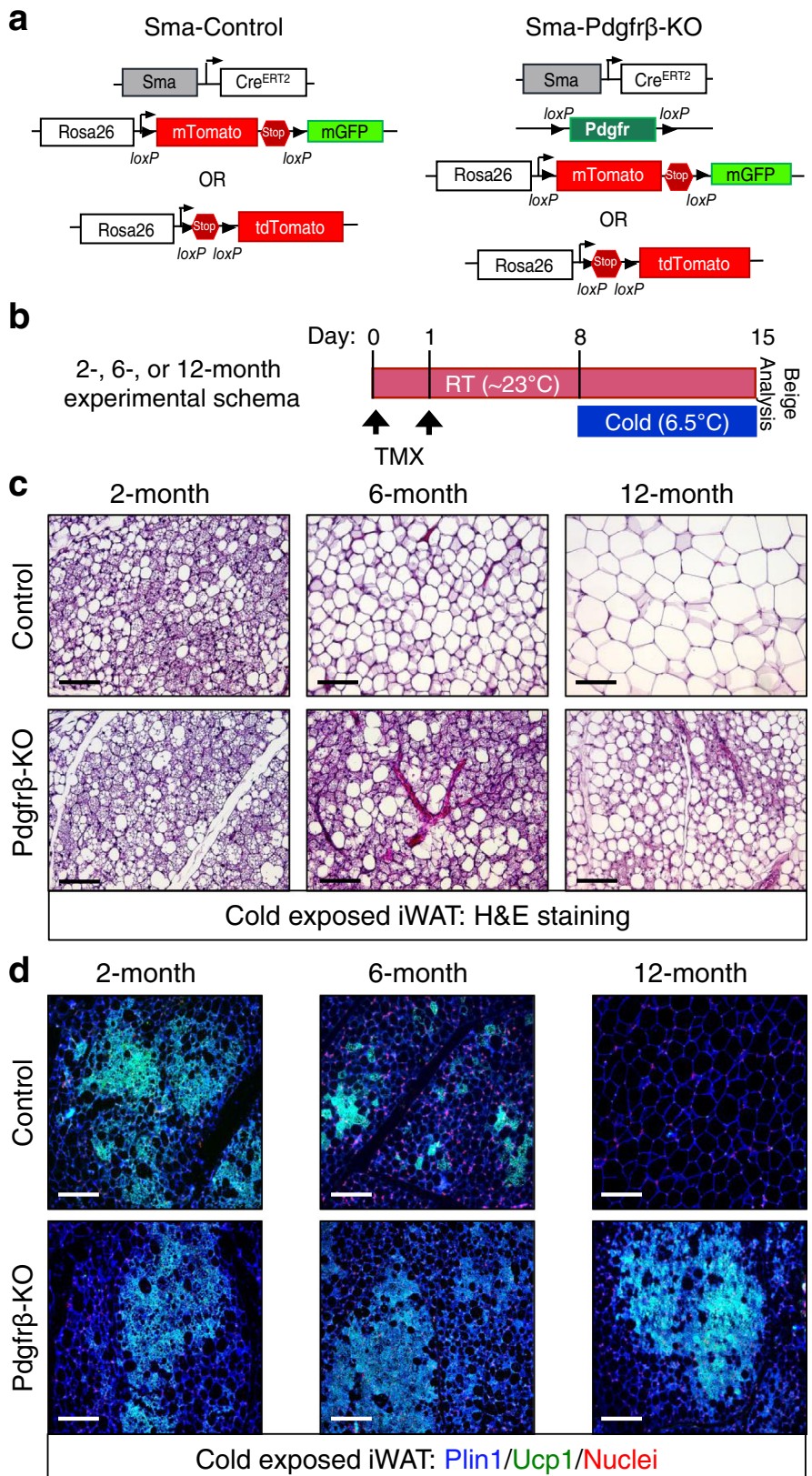

**Fig. 2 | Loss of Pdgfrβ restores beige adipocyte development in ageing mice.**
**a** Allelic schematic for generating Sma-Control and Sma-Pdgfrβ-KO mouse models. The Sma-Cre[ERT2] mouse model was combined with either R26-[tdTomato(RFP)] or R26-[mTmG] serving as the control model. The control model (*Sma-Cre[ERT2]; R26-[tdTomato(RFP)]*) was then combined with the Pdgfrβ[fl/fl] conditional mouse model to create Sma-Pdgfrβ-KO mice. In the presence of TMX, the reporter will be turned on while the Pdgfrβ allele will be excised. **b** Experimental schema: Sma-Control (Control) and Sma-

Pdgfrβ-KO (Pdgfrβ-KO) mice littermates were aged matched until 2, 6, or 12 months of age and then administered one dose of TMX for 2 consecutive days. Mice were randomized to RT or cold exposure for 7 days (*n* = 10–15 mice/group).
**c**, **d** Representative H&E staining (**c**) and Plin1 (blue) and Ucp1 (green) immunostaining (**d**) from cold exposed mice described in (**b**) (×20 magnification, scale bars 100 μm).

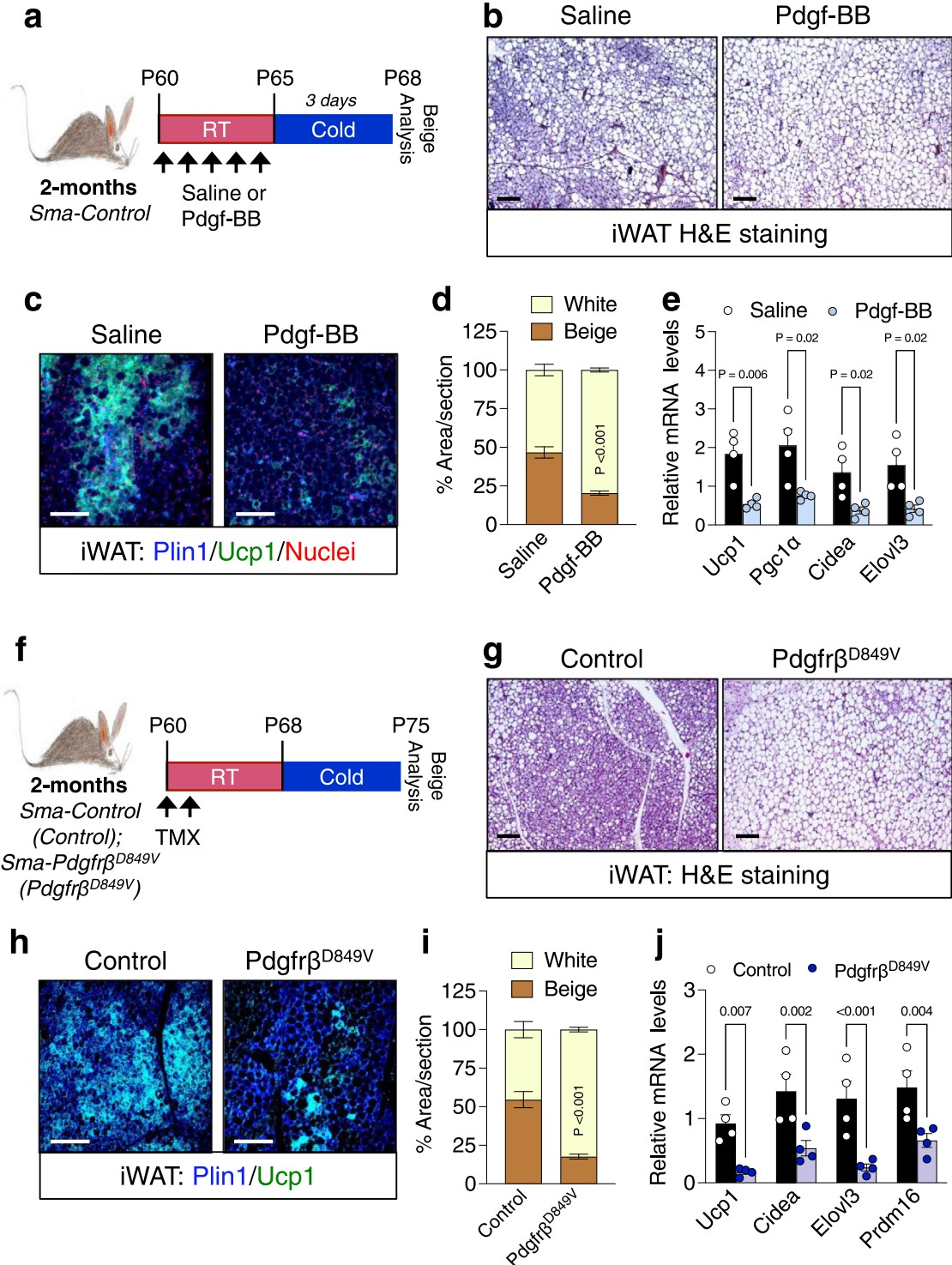

**Fig. 3 | Activating Pdgfrβ signaling blocks beige fat formation in 2-month-old mice. a** Experimental schema: 2-month-old Sma-Control male mice were administered one dose of vehicle (1X PBS) or Pdgf-BB (25 ng/mouse) for 5 consecutive days by IP injection; subsequently, cold challenged for 3 days (Vehicle $n = 8$; Pdgf-BB $n = 10$). **b** Representative H&E staining of dorsolumbar iWAT sections from cold exposed mice described in (**a**) (×10 magnification, scale bars 100 μm). **c** Representative Plin1 (blue) and Ucp1 (green) immunostaining of dorsolumbar iWAT sections from mice described in (**a**) (×20 magnification, scale bars 100 μm). **d** Quantification of beige and white adipocyte area per section ($n = 3$ images/mouse from 3 mice) from immunostained images in (**c**). **e** mRNA levels of denoted thermogenic gene expression within dorsolumbar iWAT depots from cold exposed mice described in (**a**) ($n = 4$ mice/group). **f** Experimental schema: 2-month-old TMX-induced Sma-Control and Sma-Pdgfrβ[D849V] male mice were cold temperature challenged for 7 days. **g** Representative H&E staining of dorsolumbar iWAT sections from cold exposed male mice described in (**f**) (×10 magnification, scale bars 100 μm) (Images representative of 3 independent experiments). **h** Representative Plin1 (blue) and Ucp1 (green) immunostaining from dorsolumbar iWAT sections from mice described in (**f**) (×20 magnification, scale bars 100 μm). **i** Quantification of beige and white adipocyte area per section ($n = 3$ images/mouse; 3 mice/group) from immunostained images in (**h**). **j** mRNA levels of thermogenic gene expression within dorsolumbar iWAT depots from cold exposed male mice ($n = 4$ mice/group). Data are presented as mean values ± SEM. Data were analyzed by two-tailed Student's t-test. Source data are provided within the Source data file.

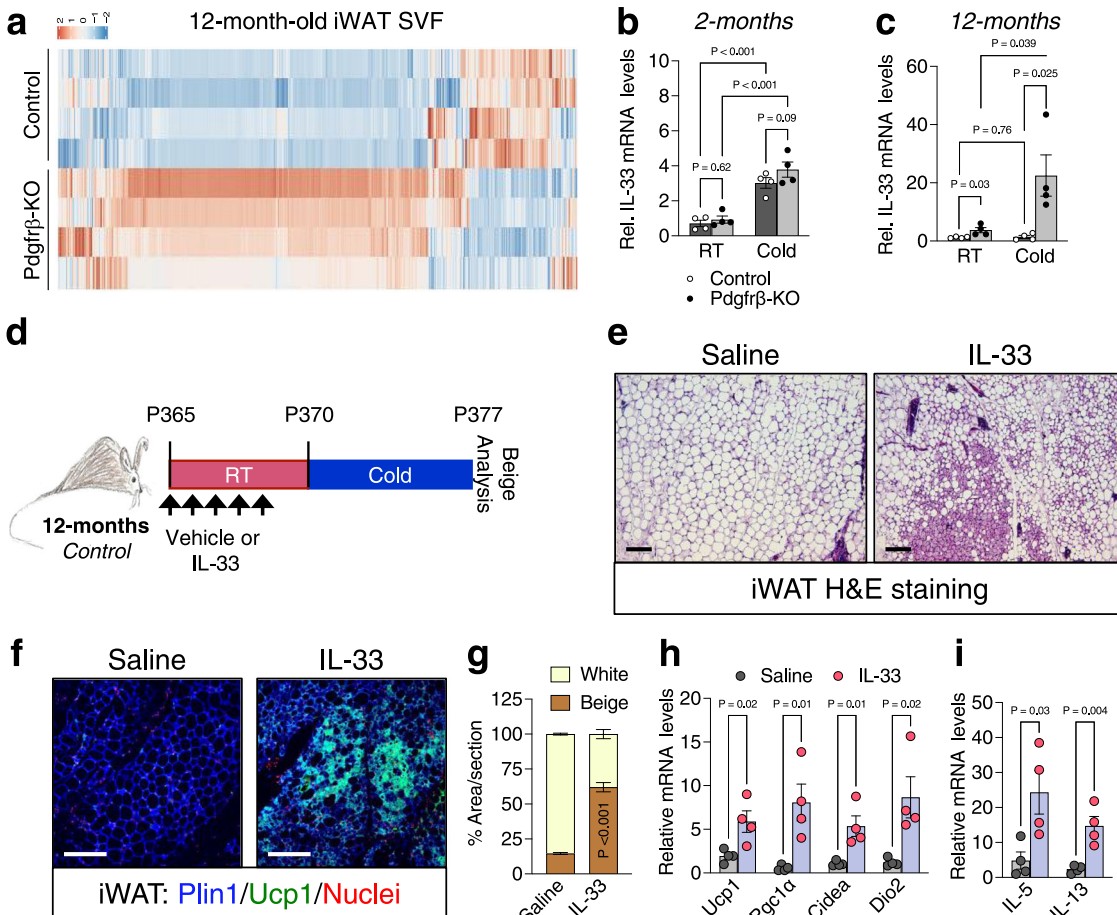

**Fig. 4 | IL-33 is mediated by Pdgfrβ signaling to regulate beige fat development.**
**a** Heatmap of gene expression profiles comparing the SVF from iWAT depots from 12-month-old Sma-Control and Sma-Pdgfrβ-KO male mice maintained at RT.
**b, c** mRNA expression of IL-33 within iWAT depots of 2- (**b**) and 12-month-old (**c**) Sma-Control and Sma-Pdgfrβ-KO mutant mice maintained at RT or cold exposed (2 days) (*n* = 4 mice/group). **d** Experimental schema: TMX-induced aged Sma-Control male mice were administered one dose of vehicle (0.1% BSA in 1x PBS) or IL-33 (12 μg/kg) for 5 consecutive days and subsequently cold exposed for 2 or 7 days. **e** Representative images of H&E staining of dorsolumbar iWAT sections from mice described in (**d**) (×10 magnification, scale bar 100 μm) (Images representative of 2 independent experiments). **f** Representative images of Plin1 (blue)

and Ucp1 (green) immunostaining of iWAT sections from mice described in (**d**) (×20 magnification, scale bar 100 μm). **g** Quantification of beige and white adipocyte area per section (*n* = 3 images/mouse; 3 mice/group) from immunostained images in (**f**). **h** mRNA levels of thermogenic gene expression within iWAT depots of cold exposed mice described in (**a**) (*n* = 4 mice/group). **i** mRNA levels of IL-13 and IL-5 within iWAT depots from 2-day cold exposed mice described in (**d**) (*n* = 4 mice/group). Data are presented as mean values ± SEM. Data was analyzed by two-tailed Student's *t*-test or two-way ANOVA for multiple comparisons. Source data are provided within the Source data file. The full list of genes and normalized counts for the gene expression analysis can be found in Supplementary Data 3.

with our previous findings[19], p38/Mapk phosphorylation was elevated within 12-month-old Sma+ APCs compared to 2-month-old APCs (Supplementary Fig. 5a). However, p38/Mapk phosphorylation within Pdgfrβ-deficient Sma+ beige APCs remained comparable to control levels, regardless of age (Supplementary Fig. 5a). We then assessed if constitutively activating Pdgfrβ resulted in enhanced p38/Mapk phosphorylation within beige APCs. However, we were unable to detect significant changes in p38/Mapk phosphorylation within Sma+ beige APCs from control and Pdgfrβ[D849V] mice (Supplementary Fig. 5b). Consistent with our *p38/Mapk* findings, gene expression of cellular senescence-inducers remained equivalent between 12-month-old control and Pdgfrβ-KO iWAT (Supplementary Fig. 5c). Moreover, cellular senescence genes were not induced within Pdgfrβ[D849V] iWAT depots (Supplementary Fig. 5d). Together, the data suggest that modulating Pdgfrβ signaling does not induce or reverse senescence signals within Sma+ beige APCs.

Our senescence tests suggest that varying Pdgfrβ signaling does not alter APC cellular senescence to influence beige fat generation. Accordingly, we revisited if the beige adipocytes generated within 12-month-old Pdgfrβ-KO iWAT depots emanated from a Sma+ cell

source[8]. To pursue this notion, we performed fate mapping analysis of iWAT sections from 12-month-old TMX-induced cold exposed control and *Pdgfrβ*-KO lineage reporter (R26-RFP) mice, as described in Fig. 2b. Immunostaining revealed that only ~9% of Ucp1+ beige adipocytes were RFP+ in *Pdgfrβ*-KO iWAT specimens, suggesting a lack of lineage tracing (Supplementary Fig. 5e, f). To ascertain if the lack of *Pdgfrβ*-KO RFP fate mapping was age dependent, we examined 2-month-old control and mutant iWAT sections for RFP reporter and Ucp1 co-localization. In contrast to 12-month-old mice, we found comparable fate mapping percentages (~50%) of Sma+ cells into Ucp1+ beige adipocytes between control and mutant samples. (Supplementary Fig. 5e, f). To continue to probe if *Pdgfrβ* deletion restored beige APC adipogenic potential, we isolated the SVF from 2- and 12-month-old control and *Pdgfrβ*-KO mutant mice. In 12-month-old cultures, neither control or mutant SVF cultures generated adipocytes (Supplementary Fig. 5g, h). However, in 2-month-old control and mutant SVF cultures had comparable in vitro beige adipogenic potential (Supplementary Fig. 5g, h). Together, these data suggest that ablating Pdgfrβ does not restore beige adipogenic potential of ageing Sma+ beige APCs rather, alternative cellular sources appear to

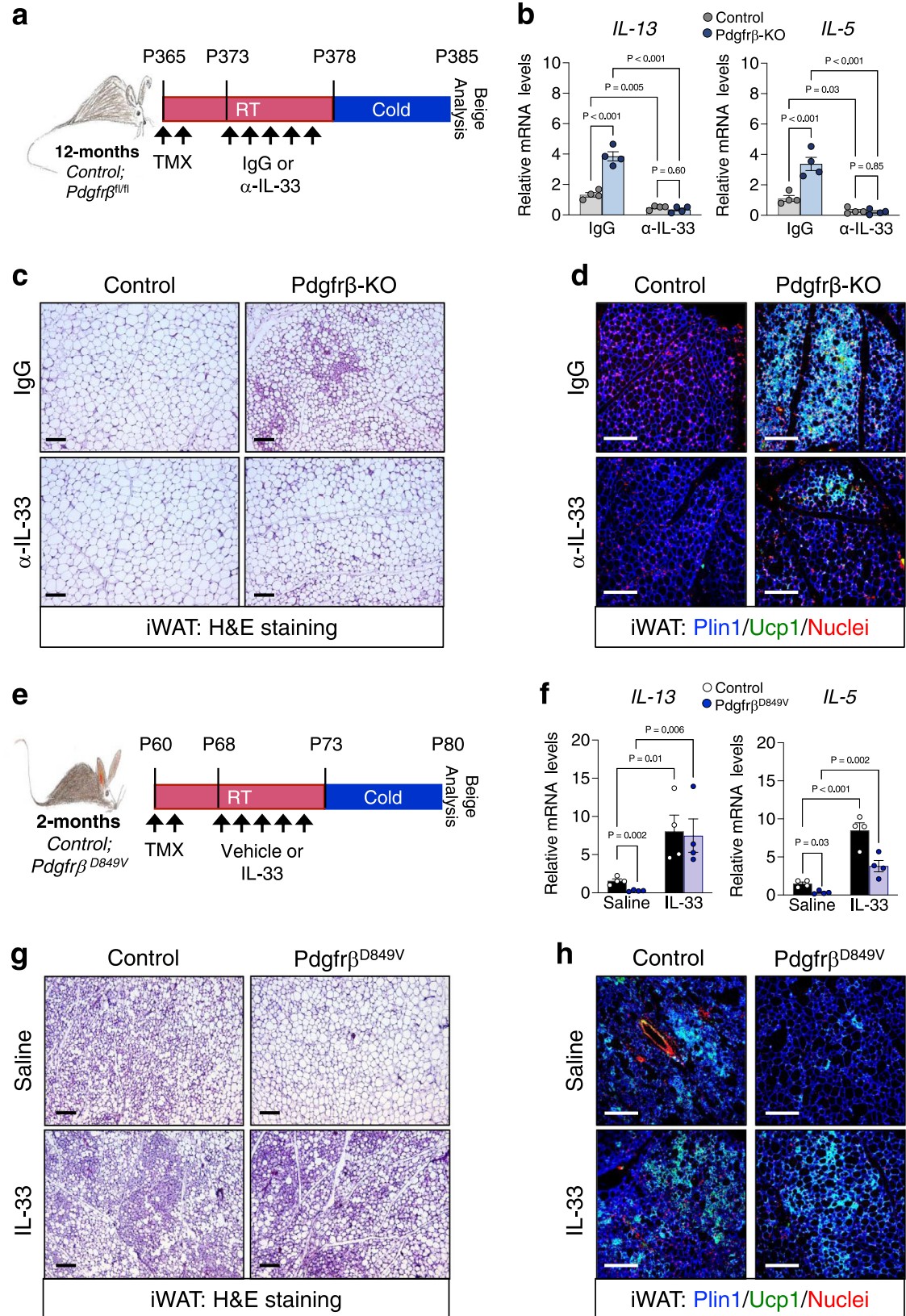

be engaged to generate cold-induced beige fat cells within ageing iWAT depots.

## Pdgfrβ activity regulates IL-33 iWAT availability

Towards delineating relevant mechanisms, we performed RNA-seq on the SVF from iWAT depots from TMX-induced 12-month-*Pdgfrβ*-KO

maintained at RT. We compared *Pdgfrβ*-KO gene expression profiles to the RNA-seq data from 12-month-old controls from Fig. 1c (Fig. 4a and Supplementary Data 3). We focused on gene expression changes that could reflect alterations in beige adipogenic potential. We found an upregulation in the beige fat inducer, *interleukin-33 (IL-33)* within *Pdgfrβ*-KO iWAT SVF[15,45] (Fig. 4b, c). Interestingly, our RNA-seq data

**Fig. 5 | Pdgfrβ-KO induced beige fat development is mediated by IL-33.**
**a** Experimental schema: TMX-induced 12-month-old Sma-Control and Sma-Pdgfrβ-KO male mice were administered one dose of mouse IgG (1 μg/mouse) or α-IL-33 (1 μg/mouse) antibodies for 5 consecutive days and subsequently cold exposed for 2 or 7 days. **b** mRNA levels of IL-13 and IL-5 within iWAT depots from 2-day cold exposed mice described in (**f**) (*n* = 4 mice/group). **c** Representative H&E staining of dorsolumbar iWAT sections from 7-day cold exposed mice described in (**a**) (×10 magnification, scale bars 100 μm). **d** Representative Plin1 (blue) and Ucp1 (green) immunostaining from iWAT sections from 7-day cold exposed mice described in (**a**) (×20 magnification, scale bars 100 μm) (Images representative of 2 independent experiments). **e** Experimental schema: 2-month-old TMX-induced Sma-Control and Sma-Pdgfrβ^D849V were administered one dose of vehicle (0.1%BSA in 1xPBS) or recombinant murine IL-33 (12 μg/kg) for 5 consecutive days and subsequently cold exposed for 2 or 7 days. **f** mRNA levels of IL-13 and IL-5 within iWAT depots from 2-day cold exposed mice described in (**e**) (*n* = 4 mice/group). **g** Representative H&E staining of dorsolumbar iWAT sections from vehicle or IL-33 treated Pdgfrβ^D849V 7-day cold exposed mice as described in (**e**) (×10 magnification, scale bars 100 μm). **h** Representative of Plin1 (blue) and Ucp1 (green) immunostaining of dorsolumbar iWAT sections from Control and Pdgfrβ^D849V 7-day cold exposed mice (×20 magnification, scale bars 100 μm) (Images representative of 2 independent experiments). Data are presented as mean values ± SEM. Data were analyzed by two-tailed Student's *t*-test or two-way ANOVA for multiple comparisons. Source data are provided within the Source data file.

comparing 2- and 12-month-old control mice showed similar *IL-33* expression levels, as previously observed[46] (from Fig. 1c) (Supplementary Fig. 6a). This data suggested that basal *IL-33* expression remains the same with age, but rather may reflect changes in the ability of cold temperatures to induce *IL-33* expression. Thus, we evaluated whether cold temperatures regulated *IL-33* expression between control and mutant mice. Of note, we used a 2-day cold exposure, a time when beige fat has not appeared[47], to capture changes in *IL-33* but in the absence of beige adipocytes. We found that ageing impaired cold temperatures to induce *IL-33* expression (Fig. 4b, c) whereas, *Pdgfrβ* deletion, reversed the cold-induced age-dependent blockade on *IL-33* expression (Fig. 4b, c). Consistent with this notion, we found that *IL-33* expression was not induced within cold exposed *Pdgfrβ^D849V* iWAT depots compared to control depots, suggesting that Pdgfrβ may regulate IL-33 availability to regulate beige fat development (Supplementary Fig. 6b).

IL-33 is a major driver of Th2-cell and group 2 innate lymphoid cell (ILC2) accrual and activation[15,17]. Moreover, IL-33 administration and ILC2 activation are sufficient to enhance beige fat development but not BAT[15,17,45]. Therefore, age-associated changes in cold-induced *IL-33* expression led us to investigate if Th2-cytokine signaling was affected by Pdgfrβ signaling. Indeed, we observed changes in *IL-5* and *IL-13*, byproducts of immune cells such as Th2, and ILC2 activation[48], within cold-exposed 12-month-old *Pdgfrβ*-KO iWAT (Supplementary Fig. 6c). In contrast, we observed lower *IL-13* and *IL-5* gene expression within iWAT depots from cold-exposed *Pdgfrβ^D849V* mice (Supplementary Fig. 6d). Together the data suggest that age-associated changes in *Pdgfrβ* expression and signaling within beige APCs can influence *IL-33* expression to facilitate changes in immune cell activity.

### IL-33 mediates *Pdgfrβ*-KO induced beiging

The above data suggested that IL-33 WAT availability is altered by *Pdgfrβ* expression; however, does IL-33 mediate cold induced-beige fat biogenesis in ageing mice and is IL-33 required for *Pdgfrβ*-KO-induced cold temperature beiging? To test if IL-33 can promote beige fat development in 12-month-old control mice, we administered one dose of IL-33 (12 μg/kg) for 5 consecutive days and subsequently, cold exposed mice for 7 days (Fig. 4d). H&E staining, Ucp1 immunostaining, and beige fat quantification revealed that IL-33 induced the generation of beige adipocytes (Fig. 4e–g). In addition, the thermogenic transcriptional program also appeared to be induced in response to IL-33 treatment (Fig. 4h). Moreover, IL-33 induced the expression of *IL-13* and *IL-5* suggesting type 2 cytokine signaling activation is involved in beige fat formation (Fig. 4i).

To probe if IL-33 is required for *Pdgfrβ*-KO-induced beige fat generation, we administered IL-33 neutralizing antibodies to 12-month-old TMX-induced aged control and mutant mice; subsequently, mice were cold exposed for 2 or 7 days (Fig. 5a)[49]. After anti-IL-33 administration and acute cold exposure, we found reduced *IL-5* and *IL-13* expression in the presence of anti-IL-33 antibodies, suggesting less immune cell activation (Fig. 5b). Regarding beige fat development,

we found that neutralizing IL-33 in Pdgfrβ-KO mice significantly reduced beige adipocyte appearance and development after 7 days of cold exposure (Fig. 5c, d and Supplementary Fig. 7a). Directed qPCR analysis of thermogenic genes confirmed our histological findings, suggesting that IL-33 mediates *Pdgfrβ*-KO induced beiging (Supplementary Fig. 7b).

To test if IL-33 could restore beige fat development in the presence of constitutively active Pdgfrβ, we administered recombinant IL-33 (12 μg/kg)[15] to 2-month-old control and *Pdgfrβ^D849V* mice for 5 consecutive days and subsequently cold exposed the mice for 2 or 7 days (Fig. 5e). IL-33 administration induced detectable differences in *IL-5* and *IL-13* gene expression in response to IL-33 administration within iWAT from both control and *Pdgfrβ^D849V* mice (Fig. 5f). Strikingly, IL-33 administration to *Pdgfrβ^D849V* mice restored beige adipocyte generation within iWAT depots compared to vehicle-treated mutant mice after 7 days of cold exposure as assessed by histological staining and gene expression (Fig. 5g, h and Supplementary Fig. 7c, d). Overall, these data appear to suggest that Pdgfrβ regulates IL-33 availability to alter beige fat development.

### Blocking Stat1 reverses age-induced beige adipogenic failure

To investigate how APC Pdgfrβ signaling suppresses IL-33 availability, we examined potential amplifiers of Pdgfrβ signaling. We focused on Stat1 (Signal Transducer and Activator of Transcription-1) due to its involvement in Pdgfrβ signaling in several disease states such as atherosclerosis and Kosaki overgrowth syndrome[50,51]. Moreover, Stat1 signaling has been shown to modulate *IL-33* expression and immune responses[52], suggesting a possible role in the age-associated decline in beige APC-immune cell communication. In addition, our RNA seq data suggested that *Stat1* was upregulated in an age-dependent manner (Supplementary Fig. 1d, e). In support of this notion, immunoblotting revealed an age-associated increase in total Stat1 within the SVF of iWAT from 2- and 12-month-old mice (Supplementary Fig. 8a). Using flow cytometry, we found that the phosphorylation status of Stat1 was elevated in 12-month-old Sma+ beige APCs compared to 2-month-old beige APCs (Fig. 6a). Moreover, Stat1 phosphorylation was reduced within 12-month-old beige APCs lacking Pdgfrβ, mirroring 2-month-old phosphorylated Stat1 levels. In contrast, constitutively activating Pdgfrβ increased Stat1 phosphorylation, mirroring the 12-month-old beige APCs (Fig. 6a and Supplementary Fig. 8b).

To investigate if blocking Stat1 phosphorylation could modulate beige adipocyte development, we administered one dose of the Stat1 inhibitor, fludarabine, an FDA approved drug to treat leukemia and lymphoma[53,54], for 5 consecutive days to 12-month-old control mice (Fig. 6b). Flow cytometric studies verified that fludarabine could reduce Stat1 phosphorylation in 12-month-old Sma+ beige APCs (Supplementary Fig. 8c). Moreover, after 7 days of cold exposure, fludarabine administration restored beige adipocyte development. Strikingly, H&E staining of iWAT sections revealed widespread "beiging" by fludarabine compared to sections from vehicle-treated mice (Fig. 6c). In agreement, fludarabine increased

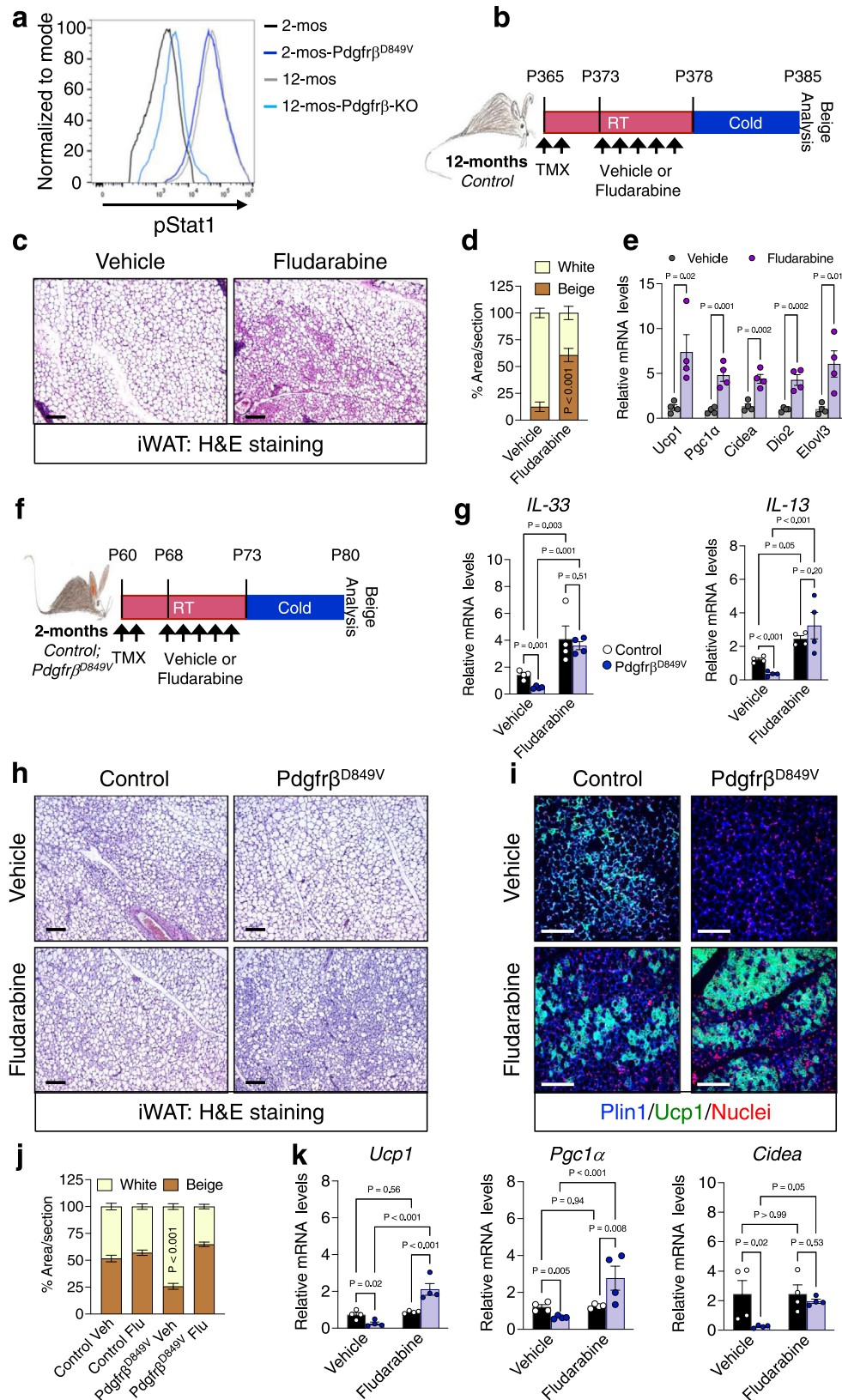

Ucp1+ immunostaining and heightened thermogenic gene expression compared to vehicle-treated specimens (Fig. 6d, e and Supplementary Fig. 8d). As a next step, fludarabine treatment also elevated *IL-33* and *IL-13* mRNA expression within iWAT depots (Supplementary Fig. 8e).

**Blocking Stat1 reverses Pdgfrβ-induced beige fat decline**

To further probe the effects of fludarabine on Pdgfrβ-Stat1 APC signaling, we posited that treating 2-month-old constitutively active *Pdgfrβ^{D849V}* mice with fludarabine would restore beige adipogenic function (Fig. 6f). Flow cytometric analysis of 2-month-old *Pdgfrβ^{D849V}*

**Fig. 6 | Stat1 phosphorylation mediates Pdgfrβ induced beige adipogenic failure. a** Sma-mGFP+ cells were FACS isolated from 2-month-old TMX-induced Sma-Control and Sma-Pdgfrβ[D849V] or from 12-month-old TMX-induced Sma-Control and Sma-Pdgfrβ-KO mice and examined for phosphorylated Stat1 by flow cytometry. **b** Experimental schema: 12-month-old TMX-induced Sma-Control mice were administered one dose of vehicle (5% DMSO) or fludarabine (3 mg/kg) for 5 consecutive days and subsequently cold challenged for 2 or 7 days. **c** Representative H&E staining of dorsolumbar iWAT sections from 7-day cold exposed mice described in (**b**) (×10 magnification, scale bars 100 μm). **d** Quantification of beige and white adipocyte area per section (n = 3 images/mouse; 3 mice/group) from immunostained images (Supplementary Fig. 8d). **e** mRNA levels of denoted thermogenic genes within dorsolumbar iWAT depots from mice described in (**b**) after 7 days of cold exposure (n = 4 mice/group). **f** Experimental schema: 2-month-old

TMX-induced Sma-Control and Sma-Pdgfrβ[D849V] male mice were administered one dose of vehicle (5% DMSO) or fludarabine (3 mg/kg) for 5 consecutive days and subsequently cold exposed for 2 or 7 days. **g** mRNA levels of IL-33 and IL-13 markers within iWAT depots from 2-day cold exposed mice described in (**f**) (n = 4 mice/group). **h, i** Representative images of H&E staining (**h**) and Plin1 (blue) and Ucp1 (green) immunostaining (**i**) of dorsolumbar iWAT sections from cold exposed mice described in (**f**) (×10 and ×20 magnification, scale bars 100 μm) (Images representative of 2 independent experiments). **j** Quantification of beige and white adipocyte area per section (n = 3 images/mouse; 3 mice/group) from immunostained images in (**i**). **k** mRNA levels of denoted thermogenic gene expression within iWAT depots from mice described in (**f**) (n = 4 mice/group). Data are presented as mean values ± SEM. Data were analyzed by two-tailed Student's t-test or two-way ANOVA for multiple comparisons. Source data are provided within the Source data file.

beige APCs revealed that fludarabine effectively reduced Stat1 phosphorylation (Supplementary Fig. 8f). Furthermore, *Pdgfrβ[D849V]* mice treated with fludarabine had an increase in *IL-33* and *IL-13* gene expression, suggesting immune cell activation (Fig. 6g). Next, we evaluated cold-induced beige fat formation in response to fludarabine. After 7 days of cold exposure, we found comparable beiging between control mice treated with either vehicle or fludarabine (Fig. 6h–j). In contrast, treating *Pdgfrβ[D849V]* mice with fludarabine significantly restored cold-induced Ucp1+ beige adipocyte generation and increased thermogenic gene expression (Fig. 6k). Overall, the data suggest that Stat1 signaling appears to facilitate Pdgfrβ-induced suppression of beige fat formation in aged mice.

### Targeting Pdgfrβ restores beige fat formation in aged mice

Towards a clinical step, we tested if pharmacologically blocking Pdgfrβ signaling could restore beige adipocyte generation in adult mice. Twelve-month-old TMX-induced *Sma*-Control male mice were randomized to vehicle, imatinib (5 mg/kg), an FDA approved drug to treat chronic myeloid leukemia[55], or SU16f (2 mg/kg), a potent and selective inhibitor of Pdgfrβ[56], for 5 days while maintained at RT and subsequently cold exposed for 7 days (Fig. 7a). Of note, both inhibitors effectively downregulated the Pdgfrβ target genes, *Axud1* and *Arid5b*[57], within iWAT depots (Supplementary Fig. 9a). In response to cold temperatures, mice treated with either imatinib or SU16f appeared to invoke a beige adipocyte phenotype as evaluated by H&E staining, Ucp1 immunostaining, and beige adipocyte quantification of iWAT sections compared to vehicle-treated mice (Fig. 7b–d and Supplementary Fig. 9b). Consistent with the notion of beige fat development, directed qPCR analysis revealed a robust thermogenic gene program induction within iWAT depots from imatinib and SU16f treated mice (Fig. 7e). Vehicle, imatinib, or SU16f treatments did not appear to alter BAT as assessed by weight, morphology, and gene expression (Supplementary Fig. 9c–e).

To assess if pharmacologically targeting Pdgfrβ was age-specific, we treated 2-month-old juvenile male mice with either one dose of vehicle, imatinib, or SU16f for 5 consecutive days and subsequently, mice were cold exposed for 7 days (Supplementary Fig. 9f). We found that vehicle and drug-treated mice appeared to have comparable levels of beige fat appearance and displayed equivalent molecular markers of beige adipocyte induction, resonating with our conditional deletion of *Pdgfrβ* in young and aged mice (Supplementary Fig. 9g–j).

We next assessed if pharmacologically blocking Pdgfrβ induced age-dependent changes in Stat1 phosphorylation and immune cell activation in 12-month-old control mice (Fig. 7a). By flow cytometric analysis, imatinib and SU16f were sufficient to reduce Stat1 levels within Sma+ beige APCs compared to vehicle treatment (Fig. 7f). Moreover, blocking Pdgfrβ signaling also resulted in elevated *IL-33* and *IL-13* mRNA levels after cold exposure (Fig. 7g). Overall, these data suggest that pharmacologically blocking Pdgfrβ restores a declining age-dependent niche circuit, by potentially allowing Sma+ APCs to

communicate with immunological cells to create a beige adipogenic niche.

## Discussion

Cold temperatures induce energy burning beige adipocytes to activate metabolism, boosting energy expenditure[10]. This attribute has and continues to attract clinical attention as a therapy to counteract obesity and metabolic disorders[10,58]. However, the inability to generate cold-induced beige fat in ageing humans has created a clinical obstacle to their therapeutic promise[12]. Thus, finding ways to restore beige adipocyte development in adult humans has clinical utility. Here, we reveal an age-dependent upregulation in *Pdgfrβ* expression and signaling which is an essential determinant of beige fat development in adult mammals. The age-dependent ramping of Pdgfrβ signaling via Stat1 appears to be independent of cellular senescence. Instead, Pdgfrβ-Stat1 signaling downregulates *IL-33* expression, impairing type 2 cytokine signaling. The concomitant loss of cold-induced immune cell activation appears to reduce beige adipogenesis. Collectively, these studies put-forth a communication network between beige APCs and immunological cells to generate beige adipocytes that deteriorates with age (Supplementary Fig. 10), potentially fostering metabolic imbalance and age-associated WAT expansion and accumulation.

Complementary functional tests demonstrate that perivascular cells communicate with the immunological niche to regulate beige adipogenic potential[59,60]. For instance, Chawla and colleagues reported that type 2 cytokine signaling results in the recruitment and activation of eosinophils and ILC2s to promote cold-induced beige adipogenesis[17,45]. In addition, Artis and colleagues also established that IL-33-dependent ILC2 recruitment and activation is necessary for beige adipogenesis via ILC2-derived methionine-enkephalin peptides[15]. In further support, Gupta and colleagues showed that Pdgfrβ marked beige APCs are required for beige fat formation and express IL-33[35,61]. Our study strengthens this notion of a beige APC- type 2 cytokine signaling communication network by identifying Pdgfrβ-Stat1 signaling as a mediator of IL-33 WAT availability. That is, hyperactivation of Pdgfrβ signaling—as observed with ageing—reduces IL-33 availability and potentially reduces type 2 cytokine signaling activation. Yet, *Pdgfrβ* deletion induced upregulation of IL-33 alone does not appear to be sufficient to restore beige fat formation and requires cold temperature stimulation. Continued examination into the full extent of Pdgfrβ signaling within APC biology and the type of immunological cells suppressed will be critical to address age-related mechanisms impairing beige fat development. Nevertheless, restoring this pathway, by blocking Pdgfrβ or Stat1 or even bypassing it by adding IL-33, showed revitalized beiging within aged mice. However, additional findings by Goldberg and colleagues revealed that WAT ILC2 immune cell stimulation may be finite. For instance, with advanced ageing, ILC2s become dysfunctional and unresponsive to IL-33 induced activation and dampens cold temperature survival[46]. More studies aimed at understanding beige adipogenic failure will provide critical clues to

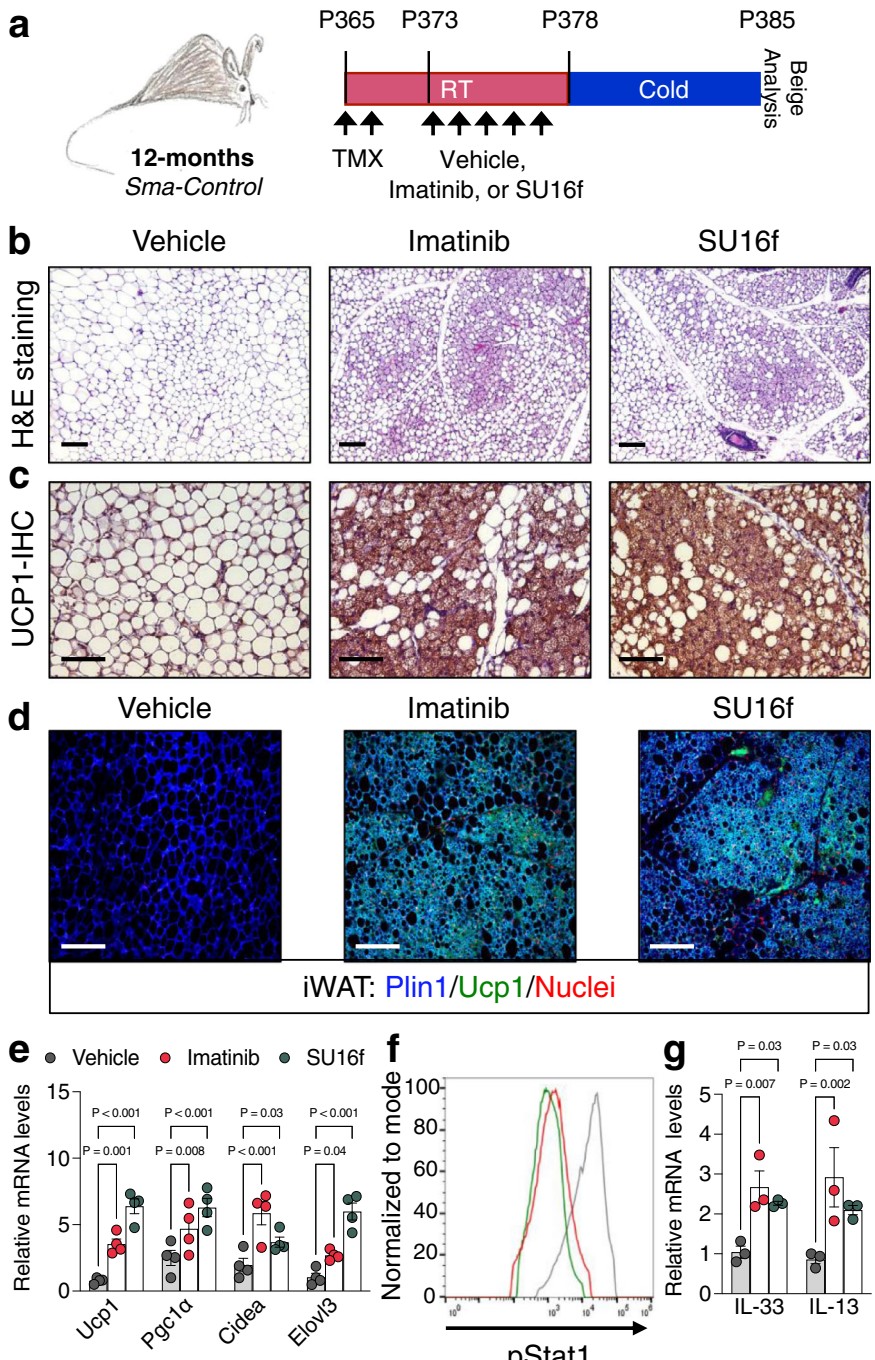

**Fig. 7 | Pharmacologically blocking Pdgfrβ restores beige adipogenesis.**
**a** Experimental schema: 12-month-old TMX-induced Sma-Control male mice
were administered one dose of vehicle (5% DMSO), imatinib (5 mg/kg), or SU16f
(2 mg/kg) for 5 consecutive days and subsequently cold challenged for 2 or
7 days (*n* = 8 mice/group). **b**–**d** Representative H&E staining (**b**), Ucp1-IHC (**c**),
and Plin1 (blue) and Ucp1 (green) immunostaining (**d**) of dorsolumbar iWAT
sections from 7-day cold exposed mice described in (**a**) (×10 or ×20 magnifi-
cation, scale bars 100 μm). **e** mRNA levels of denoted thermogenic gene
expression within dorsolumbar iWAT depots from mice described in (**a**) (*n* = 4
mice/group). **f** Sma-mGFP+ cells were FACS isolated from mice described in (**a**)
prior to cold exposure. Representative flow cytometric histogram plot of Sma+
beige APCs examined for phosphorylated Stat1. **g** mRNA levels of IL-33 and IL-13
within iWAT depots from 2-day cold exposed mice described in (**a**) (*n* = 3 mice/
group). Data are presented as mean values ± SEM. Data were analyzed by two-
tailed Student's *t*-test. Source data are provided within the Source data file.

develop new methods to restore beige fat biogenesis, especially in
older individuals and ageing populations.

Our current study supports the notion that beige APCs appear to
be senescent but suggest other age-dependent mechanisms can
interfere with the beiging process[31,62]. Our previous study supported
the notion that ageing beige APCs acquire hyperphosphorylation of
p38/MAPK, orchestrating the senescent beige APC signaling pathway[19].

Using RNA-seq, we confirmed changes in cellular aging and senescent-
like pathways that might regulate beige APC behavior and function.
Nevertheless, still at large are the factors and niche-derived mediators
driving beige APC senescence. Moreover, not only are senescent cells
present in adult WAT but also can be observed in obese adipose tissue
samples. Removal of senescent cells within WAT can restore metabolic
health and counteract obesogenic disruption[63]. Critically, obese

patients also have an impaired capability to generate cold-induced beige adipocytes and have altered ILC2 activation[15,63–65]. However, it is unclear if Pdgfrβ-Stat1 signaling facilitates obesogenic beige fat failure. Further experimentation into if and how Pdgfrβ expression and signaling becomes augmented during obesity to influence beige fat development remains unrealized. Further, additionally metabolic assessment studies will provide essential insight into Pdgfrβ-mediated metabolic flexibility in ageing. In addition, age-related transcriptional regulatory mechanisms in *Pdgfrβ* gene expression could provide further clarity into how age-associated immunological changes influence WAT health, which could provide new insight into cellular ageing and niche networks.

Cold temperatures induce the generation of beige fat cells; however, there appear to be several beige adipogenic cellular pools such as mural cells, perivascular fibroblast, MyoD+ cells, proliferating Ucp1+ cells, and interconverting adipocytes[35,66–72]. Fate mapping results in young mice support the notion that additional cellular sources, besides Sma-marked perivascular cells, can be stimulated to generate beige adipocytes. Our data also support the notion that Pdgfrβ marked cells are not a major contributor to beige adipocyte generation in young mice[35,39,73]. But this observation appears to be primarily due to *Pdgfrβ* expression which is lower in young beige APCs compared to aged APCs (Fig. 1c, d). Also, what additional cell populations can be provoked to make beige adipocytes in response to Th2 cytokine signaling remain to be fully elucidated. New genetic tools aimed at dual population indelible marking could be valuable to examine the distribution and contribution of cellular sources to beige fat biogenesis under various genetic, pharmacologic, and environmental conditions and duration. Nonetheless, our studies support a cooperation network between Sma+ beige APCs and immune cell activation. Future investigation into how these cellular communication networks change with age could be exploited to inform new strategies to target age-associated fat mass expansion and metabolic decline.

## Methods

### Mouse models
All animal experiments were performed according to procedures approved by the Cornell University Institutional Animal Care and Use Committee under the auspices of protocol number 2017-0063. *Sma-CreERT2* mouse model was generously obtained from Drs. Pierre Chambon and Daniel Metzger[40]. *Sma-CreERT2* mice were combined with either Rosa26-tdtomato (stock #007914) or Rosa26-mTmG (stock #007676) mice from Jackson Laboratories. *Sma-CreERT2* reporter mice were crossed with either Pdgfrβ D849V (stock #018435) or Pdgfrβfl/fl (stock #010977) from Jackson Laboratories. Offspring were intercrossed for six generations prior to experimentation and were maintained on mixed C57BL6/J-129SV background. To induce recombination, denoted mice were administered one dose of TMX (50 mg/kg; Cayman Chemical: 13258) dissolved in sunflower seed oil (Sigma, item no: S5007) for 2 consecutive days via intraperitoneal (IP) injection. After the final TMX injection, mice were maintained for 7 days at room temperature prior to experimentation, as a TMX washout period. For cold temperature exposure, mice were housed in a 6.5 °C cold chamber (Power Scientific RIS70SD) or mice were maintained at RT (~23–25 °C; ~35% humidity). Mice were maintained in a vivarium on a 14:10-h light/dark cycle with free access to food (Chow diet: Teklad LM-485 Mouse/Rat Sterilizable Diet) and water. All animal experiments were performed on male mice at denoted ages per experiment. All experiments were performed on 3 or more mice per cohort and performed at least twice. Animals were euthanized by carbon dioxide asphyxiation and cervical dislocation was performed as a secondary euthanasia procedure.

**Physiological measurements.** Temperature was monitored daily (~18:00 EST) using a TH-5 Thermalert Clinical Thermometer (Physitemp) attached to a RET-3 rectal probe for mice (Physitemp). The probe was lubricated with glycerol and was inserted 1.27 centimeters (1/2 inch) and temperature was measured when stabilized.

**Pharmacological administration of imatinib, SU16f, Pdgf-BB, IL-33, anti-IL-33 antibodies, and fludarabine.** Mice at denoted ages were administered one dose of vehicle (5% DMSO), imatinib (5 mg/kg; Sigma: SML1027) or SU16f (2 mg/kg; Tocris Bioscience: 3304) dissolved in 5% DMSO 1XPBS solution for 5 consecutive days via IP injection. Recombinant Pdgf-BB or IL-33 were administered to young mice at one dose of vehicle (1X PBS), Pdgf-BB (25 ng/mouse; Pepro-Tech: 315-18), or IL-33 (12 μg/kg; PeproTech 210-33) for 5 consecutive days via IP injection. For anti-IL-33 neutralizing antibody experiments, TMX-induced 12-month-old mice were administered one dose of mouse IgG control (1 μg/mouse; Invitrogen 31903) or anti-IL-33 antibody (1 μg/mouse; R&D systems AF3626) by IP injection for 5 consecutive days. Subsequently, mice were cold temperature challenged. For fludarabine treatments, mice at denoted ages were randomized to one dose of vehicle (5% DMSO) or fludarabine (3 mg/kg; Tocris Bioscience: 3495) for 5 consecutive days via IP injection.

### Human adipose SV samples
Isolated human adipose stromal vascular cells were purchased from Zen-Bio (Research Triangle Park). Cells were isolated from the abdomen and hip of young (23.5 ± 3.87 years) and old (49.33 ± 6.02 years), non-obese BMI (Young 23.43 ± 4.32, Aged 24.67 ± 2.61) female patients[19]. Cells from four patients/group were cultured in DMEM/F12 supplemented with 10% FBS and were only passaged once prior to use. Four cell lines from each group were used, and data are expressed as means ± SEM.

### Adipose SV cell isolation
Fat pads were removed and a pair of inguinal adipose depots from a single mouse were minced and placed in 10 ml of isolation buffer (0.1 M HEPES, 0.12 M NaCl, 50 mM KCl, 5 mM D-glucose, 1.5% BSA, 1 mM CaCl₂) supplemented with collagenase type I (10,000 units: LS004194) and incubated in a 37 °C incubator with gentle agitation for ~1 h[74]. Serum-free Dulbecco's modified Eagle's medium Nutrient Mixture F-12 Ham (Sigma, cat. no. D8900 and D6421) (DMEM/F12) media was added to the digested tissue and strained through a 100 μm cell strainer. Samples were spun at 200 × g for 10 min. The supernatant was removed, and the pellet was resuspended in 10 ml of erythrocyte lysis buffer (155 mM NH₄Cl, 10 mM KHCO₃, 0.1 mM EDTA). After a 5-min incubation period, growth media (DMEM/F12 supplemented with 10% fetal bovine serum (FBS)) was added, mixed, and strained through a 40 μm cell strainer. Samples were then spun at 200 × g for 5 min. Supernatant was removed and the cell pellet was resuspended in growth media and plated. After 12 h, growth media was removed and replenished.

### Adipogenesis
Isolated SV cells were grown to confluency. To induce beige adipogenesis, confluent cells were treated with beige adipogenic media one (DMEM/F12 supplemented with 5% FBS 10 μg/ml insulin, 1 μM Dexamethasone, 250 μM 3-Isobutyl-1-methylxanthine, 2 nM Triiodothyronine (T3); 250 nM Indomethycin) for 72 h. After 72 h, beige adipogenic media one was removed and replaced with beige adipogenic media two (DMEM/F12 supplemented with 5% FBS 10 μg/ml insulin, 2 nM Triiodothyronine (T3); 250 nM indomethacin). Adipogenesis was assessed by LipidTox™ staining and mRNA expression.

### Lipid staining
At the end of differentiation, media was aspirated, and adipocytes were fixed with 4% paraformaldehyde for 45 min. Adipocytes were washed thrice with 1X TBS for 5 min. Adipocytes were permeabilized using

0.3%TritonX-100 in 1X TBS for 30 min. Adipocytes were washed thrice with 1X TBS for 5 min/wash. Adipocytes were incubated with HSC LipidTox-deep red (1/1000 in 1X TBS) for 45 min rocking in the dark. Adipocytes were washed twice with 1X TBS for 5 min/wash. Adipocytes were then stained with Hoechst (1 μg/ml in 1X TBS) for 10 min. Adipocytes were washed twice with 1X TBS for 5 min/wash. Fluorescent images were collected on a Leica DMi8 inverted microscope system.

## Flow cytometry

The iWAT SVF was isolated as above and resuspended in 1X PBS along with blue fluorescent reactive dye. Cells were then pelleted (1200 rpm for 10 min), resuspended in 0.3–0.5 ml of FACS buffer (2.5% horse serum; 2 mM EDTA in 1X PBS with 1X protease/phosphatase inhibitor cocktail) and filtered through a 5 ml cell-strainer capped FACS tube (BD Falcon). Cell sorting was performed on BD Biosciences FACSAria Fusion, or cells were analyzed on a Thermo-Fisher Attune NxT cytometry. Viable cells were gated from the blue fluorescent reactive dye negative population followed by singlet forward and side scatter pattern and mGFP+ viable cells were sorted. For assessing recombination efficiency, cells were stained for alpha- Smooth Muscle Actin (NBP2-34760V) and analyzed for GFP overlap. For phosphorylated Stat1 and total Pdgfrβ analysis, mGFP+ cells were fixed with 4% PFA for 1 h at room temperature. Subsequently, cells were washed with 1X TBS and permeabilized for 30 min at room temperature with 0.3%TritonX-100 in 1X TBS. Cells were blocked with 5% donkey serum in 1X TBS for 30-min and incubated with primary antibody in 1X TBS with 5% donkey serum overnight at 4 °C. Primary antibodies used were phosphorylated Stat1 (1:200; 9167S Cell Signaling) or total Pdgfrβ (1:200; 4564S Cell Signaling). After washing, secondary antibodies were applied for 2 h at room temperature in the dark, then analyzed. Seconday antibodies used were Cy5 donkey anti-rabbit (1:200; Jackson ImmunoResearch).

## RNA isolation and qPCR

For tissues, ~200 mg of brown adipose tissue or dorsolumbar iWAT, from one mouse, was placed into Precellys tubes containing ceramic beads and 1 ml of TRIzol (Ambion 15596019). Tissues were homogenized in a Precellys 24 homogenizer using the following the settings: 3 pulses at 4500 rpm for 30 seconds with a 30 s rest between pulses with a final rest of 4 min. For cells, TRIzol was directly added to culture dishes and mechanically disrupted. RNA was extracted using the standard chloroform extraction and isopropanol precipitation method. RNA concentrations and quality were determined using a TECAN infinite F-nano+ spectrophotometer. 1 μg of RNA was converted to cDNA using the high-capacity RNA to cDNA kit (Life Technologies #4368813). For qPCR, cDNA was diluted 1:10 and added to PowerUp™ SYBR™ Green Master Mix (Life Technologies A25742) along with denoted primers. qPCR analysis was performed on an Applied Biosystems QuantStudio™ 3 Real-Time PCR System using the ΔΔ-CT method compared to the internal control, Rn18s. Data points represent a biological replicate (single mouse or culture). Each data point was performed in a technical quadruplet. qPCR primer sequences can be found in Supplementary Table 1.

## RNA sequencing

The RNAseq project was managed by the Transcriptional Regulation and Expression (TREx) Facility at Cornell University. RNA from the inguinal fat depot was isolated as previously described and submitted to the TREx Facility at Cornell for quality control analysis to determine concentration, chemical purity (Nanodrop), and RNA integrity (Agilent Fragment Analyzer). PolyA+RNA was isolated with the NEBNextPoly(A) mRNA Magnetic Isolation Module (New England Biolabs). 750 ng per sample was used for Directional RNAseq library preparation using the NEBNext Ultra II RNA Library Prep Kit (New England Biolabs) and quantified with a Qubit (dsDNA HS kit; Thermo Fisher). Libraries were then sequenced on an Illumina instrument and 20 M reads generated per library. Basic analysis including preprocessing, mapping to reference genome, and gene expression analysis to generate normalized counts and statistical analysis of differential gene expression was performed by the TREx Facility. Additional analysis including heat map generation and KEGG pathway analysis was performed using Rstudio software.

## Histological analysis

Tissues were dissected and immediately placed in 10% formalin (neutralized with 1X PBS) for 24 h. Tissues were processed using Thermo Scientific™ STP 120 Spin Tissue Processor with the following conditions: Bucket 1: 50% ethanol (45 min); Bucket 2: 70% ethanol (45 min) Bucket 3: 80% ethanol (45 min); Bucket 4 and 5: 95% ethanol (45 min); Bucket 6 and 7: 100% ethanol (45 min); Bucket 8-10: Xylene Substitute (45 min); Bucket 11 and 12: paraffin (4 h each). Tissues were embedded into cassettes using a Histostar™ embedding station. Blocks were refrigerated for at least 24 h prior to sectioning. 8–12-micron tissue sections were generated using a HM-325 microtome using low profile blades. Sections were placed in a 40 °C water bath and positioned on microscope slides. Slides were then baked overnight at 55 °C prior to staining.

**Hematoxylin and eosin (H&E) staining.** Slides were rehydrated using the following protocol: xylene (3 min 3X), 100% reagent alcohol (1 min 2X); 95% reagent alcohol (1 min 2X); water (1 min). Slides were stained in hematoxylin and eosin (H&E) staining for 2 min 30 sec and 10 repeated submerges, respectively. Slides were dehydrated in the reverse order of rehydration steps. Coverslips were mounted with Cytoseal 60 mounting media. Brightfield images were acquired using a Leica DMi8 inverted microscope system.

**Immunohistochemistry (IHC) + immunofluorescent staining.** Slides were rehydrated as previously described. Antigen retrieval was performed using 1X Citrate Buffer, made from 10X stock (Electron Microscopy Sciences R-Buffer A, 10X, pH 6; Catalog: 62706-10), and placed into antigen retriever pressure cooker (EMS Catalog #62706) for 2 h. Slides were permeabilized using 0.3%TritonX-100 in 1X TBS for 30 min and washed thrice in 1xTBS. Slides were blocked with 5% donkey serum in 1X TBS for 30-min and incubated with primary antibody in 1X TBS with 5% donkey/goat serum for either 1 h at RT (21–23 °C) or overnight at 4 °C. The following primary antibodies were used: goat anti-perilipin (1/100; Abcam: ab61682); rabbit anti-Ucp1 (1/200; Abcam: ab10983); mouse anti-DsRed (1/200; Takara: 632392); mouse anti-alpha-smooth muscle actin (1/200; NovusBio 2-34760 V). Secondaries from Jackson ImmunoResearch or Invitrogen were all used at 1/200 dilutions and incubated for 2 h at room temperature. The following secondary antibodies were used: Cy5 donkey anti-goat, Cy5 donkey anti-rabbit, Cy3 donkey anti-rabbit, or 488 donkey anti-mouse. Slides were washed and stained with Hoechst (H3570; Life Technologies) (1 μg/ml in 1X TBS) for 10 min and cover slipped with Thermo Scientific™ Shandon™ Immu-Mount™ mounting media. Fluorescent images were collected on a Leica DMi8 inverted microscope system.

For IHC, peroxidase blocking was performed after primary antibody incubation using 0.3% hydrogen peroxide solution for 10 min at room temperature. Slides were washed 3X for 2 min with 0.05% PBS-Tween 20 and incubated in biotinylated secondary antibody (Vectastain Universal Quick Kit: PK-8800). After another wash, slides were exposed to ABC-Peroxidase Solution for 30 min at room temperature (Vectastain ABC Kit: PK-4000) and washed. Slides were then incubated in peroxidase substrate solution (Peroxidase Substrate Kit Vectastain: SK-4100) and washed then counterstained with hematoxylin for 3 min. Tissue sections were then rinsed with running water for 2–5 min, dehydrated in 95% ethanol (1 min) and 100% ethanol (3 min 2X), and cleared in xylene (5 min 2X). Slides were mounted in Cryoseal 60

mounting media and brightfield images were collected under a Leica DMi8 inverted microscope.

## Beige area quantification

Three 20X immunofluorescent images containing the individual fluorescent channel for Ucp1 from 3 samples/group were quantified utilizing ImageJ Fiji software (3 images/mouse and 3 mice/group were analyzed). Images were converted to the appropriate color threshold and highlighted to determine percent of Ucp1+ fluorescence over total fluorescent signal. Output values were then corrected for background signal and averaged in their respective groups.

## Immunoblotting blot analysis

The iWAT SVF was isolated as above and lysed using 200 μL RIPA Lysis Buffer. In designated experiments, the SVF (prior to RIPA buffer addition) was additionally treated with either vehicle (1X PBS 0.1% BSA) or Pdgf-BB (25 ng/sample) for 15 min in serum-free media, collected and lysed. Lysed samples were then incubated for 30 min on ice, spun at 14,000 G for 15 min at 4 °C, and the supernatant collected. Standard curve and respective protein concentrations for samples were determined and calculated utilizing the protocol provided in the Pierce protein assay kit (Pierce™ BCA Protein Assay - ThermoScientific) and a TECAN infinite F-nano+ spectrophotometer to read absorbance. For sample preparation, the calculated volume of 100 μg of protein per sample was mixed at a 1:1 ratio of 2X SDS/DTT, then heated for 10 min at 100 °C. Prepared samples and protein ladder (Biorad 161-0374) were loaded into a 10% separating and stacking gel (Biorad 4561034). The gel was placed into a Mini-PROTEAN Tetra Electrophoresis Cell chamber (Biorad 1658004) suspended in 1x running buffer (Biorad 1610744) and ran at 90 V for ~2.5 h. Protein was then transferred onto an immobilon PSQ PVDF membrane (Millipore ISEQ0005) in 1X transfer buffer (Biorad 1610771) for 1 h at 100 V on ice. The resulting membrane was removed and washed with 1x TBS-0.1% Tween20 (TBS-T) thrice for 5 min on a rocker at RT. The membrane was then blocked with 5% BSA in 1X TBS-T for 1 h at RT and washed as previously described. Prior to primary antibody addition, the membrane was cut depending on protein molecular weight. Primary antibody in 5% BSA in 1X TBS-T was added at 4 °C, rocking overnight. The primary antibodies used are as follows: rabbit anti-Pdgfrβ (1:1000; Cell Signaling: 4564S); rabbit anti-phosphorylated Pdgfrβ (Y1009) (1:1000; Cell Signaling: 3124S); rabbit anti-Stat1 (1:1000; Millipore 06-501); rabbit anti-phosphorylated Stat1 (1:1000; Invitrogen 33-3400); rabbit anti-GAPDH (1:1000; Cell Signaling: 2118); rabbit anti-beta-tubulin (1:1000; Cell Signaling: 15155S). Membranes were washed again and submerged in secondary antibody for 2 h at room temperature, rocking (1:10,000; ThermoFisher Scientific: donkey anti-rabbit IgG (H + L) Cross-Adsorbed HRP 31458 in 5% BSA 1x TBS-T or 1:5000: Thermo-Fisher Scientific: donkey anti-rabbit IgG (H + L) Cross-Adsorbed HRP 31458 in 5% Milk 1x TBS-T. Subsequently, membranes were washed as previously described and submerged in a 1:1 solution of SuperSignal™ West Pico PLUS Chemiluminescent Substrate (ThermoScientific: 34580) for 3 min and developed utilizing a FlourChem E system (bio-techne® proteinsimple). All uncropped images of blots can be found in the source data files.

## Quantification and statistical analysis

Statistical significance was assessed by two-tailed Student's t-test for two-group comparisons. Two-way ANOVA followed was used for multiple group comparisons. Individual data points (sample size) are presented and plotted as means. Error bars are expressed as ± SEM. $P < 0.03$ was considered significant in all the experiments. The statistical parameters and the number of mice used per experiment are found in the figure legends. Mouse experiments were performed in biological duplicate or triplicate with at least three mice per group. Cell culture experiments were collected from three or four independent cultures for each sample. The flow cytometric analysis software, FlowJo version 10.8.1, BD FACSDiva Software version 9.4, and Attune Cytometric Software 5.3.2415.0 was used to analyze cell populations and antibody staining. NIH Fiji ImageJ software was used to quantify co-localization, three random fields were assessed from at least three mice/cohort. The Leica Application Suite X Microscope software was used for image acquisition and analyses. Representative H&E staining and immunochemical images were obtained from at least 4–5 replicates per group. RStudio was used for RNAseq statistical analysis, heat map generation, and KEGG pathway analysis. All graphs and statical analysis were performed using GraphPad Prism 7-9 software. Excel was used for raw data collection, analysis, and quantification.

## Reporting summary

Further information on research design is available in the Nature Portfolio Reporting Summary linked to this article.

## Data availability

The data that support the findings of this study are available in the source data provided with this paper. The RNA sequencing data that supports the findings of this paper have been deposited in the GSA database under accession code CRA015560. The molecular signature databases used can be found on the GSEAMSigDB website (https://www.gsea-msigdb.org/gsea/msigdb/index.jsp). Source data are provided with this paper.

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

## Acknowledgements

The RNAseq project was managed by the Transcriptional Regulation and Expression (TREx) Facility at Cornell University. The authors thank Sean McCabe for image analysis and quantification. The authors thank Dr. Tolunay Beker Aydemir for critically reading the manuscript and helpful suggestions. The authors thank Heather Roman for breeding the mouse colony, technical assistance, and aid in collecting preliminary observations. The authors thank the Cornell Biotechnology Resources Center Flow Cytometric Core Facility and the Center of Animal Resources and Education for excellent assistance with experimental collection and mouse husbandry, respectively. Y.J. was supported by K01 DK111771. This work was supported by the American Federation of Aging Research (AFAR), and NIH-NIDDK awards K01 DK109027, R03 DK122193, and R01 DK132264-01 to D.C.B.

## Author contributions

Y.J. and D.C.B. initiated the study. A.M.B., D.L., and D.C.B. designed the experiments and wrote and revised the manuscript. A.M.B., B.M.S., D.L., S.X., Y.J., and D.C.B. performed experiments and analyzed the results.

## Competing interests

The authors declare no competing interests.
