## [Peer Review File · Nature Communications]

Age-dependent Pdgfr β signaling drives adipocyte progenitor dysfunction to alter the beige adipogenic nicheREVIEWER COMMENTS

Reviewer #1 (Remarks to the Author):

In this manuscript, the authors investigate the effect of elevated age, at 6 and 12 months, on browning capacity in white adipose tissue in mice. They argue that loss of browning capacity may contribute to impaired maintenance of cold tolerance and metabolic health and identify a potential role for PDGFRB signaling and show that this effect is potentially mediated by ILC2 upon release of IL33. Overall, this is a conceptually intriguing analysis that, if verified would contribute a novel and important mechanism to help explain loss of brown adipose thermogenesis during aging. The wide array of datasets is a clear strength and the authors present a generally promising combination of mouse models to provide some support to their hypothesis which is worth pursuing. However, I have a number of reservations regarding this study that in my view would need to be addressed rather extensively to make this a fully convincing analysis. Specifically, I believe some conclusions are not entirely supported by data and some of the data representations are in view not ideal to allow readers to appreciate the data and their implications fully. Additionally, the paper could benefit from some careful editing and checking, several extended figures are mixed up, descriptions are sometimes slightly cursory and do not allow full appreciation of the data. Here are a few items that the authors could consider:

- In general terms, only beigeing is assessed in the study, but the bulk of thermogenesis (in mice) occurs in the regular brown fat depots, dwarfing the contribution of beige adipocytes by several orders of magnitude. The analysis of the main BAT depots, such as interscapular BAT is rather cursory, what happens to the energy metabolism contribution and thermogenesis in these depots?

- Data on cell fate mapping: While these data are certainly of interest, I believe the phenotypical data are more so. The manuscript is already rather complex and the authors could consider whether these analyses are an essential part of the manuscript. In my mind, a more thorough analysis of the different mouse models metabolic phenotypes would be rather more critical. This is true for two reasons: 1, given that most publications agree that, while browning of WAT is certainly a response to prolonged cold exposure, its contribution to overall energy expenditure is compared to true BAT depots is rather very minor in mice; and 2: most of the metabolic data are not entirely consistent and outcomes vary between the different mouse models. For example, while fludarabine quite markedly improves cold tolerance, no effect on metabolic health is observed.

- Line 85, and using relatively young animals as a model of aging throughout the manuscript: Conceptually, examining animals at 6 months and in some cases up to an age of 12 months is not really aging research in mice and one can hardly claim rejuvenation of 6-months old mice that are still fully fertile and under normal circumstances even completely healthy. While I agree that the notion of what age defines aging may be somewhat debatable, onset of senescence certainly does not occur at such early ages and I believe the analysis do not convincingly allow for such conclusions. Experimentally, at the very least a time course analysis of animals aged up to 24 months, which is often a standard for aging research on mice, should be conducted to determine what happens to browning capacity in the post-fertile period of the animal's life. Specifically, expression of Pdgfrb should be examined at later stages of life to be correlated with functional markers of browning (ideally also in humans, as 49.33 years is hardly considered very old in people). If such analyses are not possible I very recommend to remove statements that the concept of aging / rejuvenation is addressed from the manuscript.

- Figure 1: I am not sure I understand the contradicting observations described in the figure: While cold exposure reduces PDGF-BB levels, aging does too. Both would suggest reduced signaling activity through this molecule in both conditions (cold, aging). However, the deletion of the receptor enhances beigeing/ metabolic activity, which does not fit the changes in the level of the ligand.

- Figure 1 and extended figures 3 and 4: The way the data are presented for the 6-months group is rather difficult to put into context with the 2-months cohort (extended Figure 4) and the 12-months group (extended Figure 3) as data are presented in various styles, colors and combinations. In order

to avoid misleading interpretations, the authors need to present the data in a consistent way, meaning that temperatures, body weights, adiposity calculations, and glucose levels, etc., should all be reported in the same way and style, with the same axis and statistical comparisons. Related to this, the authors state in lines 99-101 that all metabolic parameters are improved in mutant mice, which is not actually what is shown: Animals of both genotypes show the expected responses to cold. The correct statistical comparison here should be between the genotypes, and here no differences are observed, meaning that there is no effect of the genotype. In summary, I find the presentation of the data somewhat misleading as this comparison is not actually made (or shown) in any of the panels 1g-1i. Moreover, for all age groups, the authors sometimes show the non-cold-exposed control groups, but not for all datasets. Why is this? Temperature data seem to be available for all three ages including the respective control groups, so I think the rest of the data should also be shown. Related to this: Quantifications of browning of multilocular iWAT or Ucp1+ iWAT or lipid droplet size in the different age groups and genotypes seems to be missing. The study hinges on differences in browning, so I think a formal quantification of the histological analyses that are shown to support this statement is warranted.

- Extended Figure 2: Data of a time course of body temperature in female mice is shown, but only one time point (final) in males in the main figure 1. What about the other time points in the male mice of the main figure 1? Also, the blood glucose levels in females seem the same in both genotypes after cold, while beige markers are increased. Wouldn't this argue somewhat against a role for *Pdgfrb* in metabolic health in female mice? The authors should assess circulating lipids in both sexes as also the fat depots are smaller, which would point towards increased lipid mobilization. There are several possible ways of addressing this, for instance: A minimum would in my view include analysis of HSL phosphorylation and ATGL levels in WAT, measurement of free fatty acids and triglycerides in circulation, and better still would be indirect calorimetry of the animals to determine whether certain nutrients are preferentially metabolized in the KO mice in cold. This would be important to confirm that indeed increased metabolic turnover occurs as a result of increased browning of WAT.

- Figure 1j: What happens to expression of *Pdgfrb* at this later time point. In theory, if beige adipocytes are formed better in the KO model, wouldn't one expect to see more adipocytes with deletion of the gene due to the induction. During the six-months phase after induction one would expect a certain level of turnover of adipocytes.

- Extended Data Fig. 4, lines 122-133: The temperature data in the young mice are very confusing. It looks like young, two-months-old, mice of both genotypes are unable to defend their body temperature during a cold challenge, as body temperature decreases in both genotypes after cold (Extended Figure 4b). The same seems to be true for 6-months old mice. Then suddenly mutant mice aged 12 months (main Figure 1f) gain the ability to defend their body temperature? These data in my view are difficult to reconcile with the concept that the authors propose in their study. To make a convincing argument here, the body temperature data of all age groups and all genotypes with and without cold exposure need to be compared among each other. It looks like the drop in young mice might be less pronounced but this would require a very careful statistical analysis of all data and housing conditions, time of day of measurements, etc. This would need to be documented very thoroughly in order to be able to determine whether such comparisons, if they were not done all at the same time, are indeed permissible.

- Figure 2/ extended figure 5: I am a bit doubtful about the specificity of imatinib. While the authors cite a 1996-paper on the ability of the substance class to inhibit PDGF receptors, I believe more recent studies have established that it blocks a number of enzymes with tyrosine kinase activity. At the very least, an analysis of potential target signaling pathways is needed. This includes known targets, i.e. *Pdgfrb* signaling (this study), *bcr-abl* signaling (for instance PMID 21098337) and *c-Kit* (PMID 17607922). As a side note, authors should re-evaluate their statements: They mention that only blood glucose is also reduced in the young cohort when in fact, the only white fat depots to be reduced in size (IGW) is equally affected at both ages, overall somewhat contradicting the statement in lines 154-155.

- Figure 3: Similar to my previous comment, the authors need to show PDGF-BB and overexpression of the constitutively active receptor act on iWAT and alter PDGF BB signaling to establish a direct effect within the tissue.

- Figure 5: What are IL33 levels in BAT and iWAT of 6- and 12-months old mice compared to younger mice and compared to circulating levels (if any circulates)? In order to link changes of Pdgfrb to IL33 it would in my view be important to investigate the entire signaling cascade. It would also help strengthen the notion that an aging-like phenotype may already occur in relatively young mice. I am surprised by the observation that mice with constitutively active PDGFRB show no differences in ILC2s when this same model features a very pronounced cold intolerance as shown in figure 3. How do the authors explain the discrepancy?

- Figure 6: I am unsure about the analysis of Stat1 phosphorylation: A thorough analysis would require assessment of basal / non-phosphorylated as well as phosphorylated forms of the Stat1 protein, for instance by western blot or staining of sorted cells on a cover slip. This links to a previous comment: The authors would need to better establish altered PDGF signaling in their models. As they have also used isolated SVF from the animal models I believe it should be feasible to more clearly assess this in an in vitro cell culture system with inducible deletion/ overexpression or recombinant PDGF-BB. Besides I can't seem to find a methods description of the flow cytometry analysis nor any of the controls used to verify antibody specificity. Intracellular stainings are rather challenging.

Minor

- Line 70: Typo, I think it is Acta2, not Acat2

- Figure 1i: What is "% delta WAT"? It is only defined as "adiposity" in the figure legend. Please provide more details on how this was calculated and what measurements or data were used for this. Are these data based on body composition analyses by NMR/DEXA? If so, the two main fractions, i.e. fat mass, lean/muscle mass) should be reported. Overall, this type of body composition analysis would be a key element of a convincing metabolic characterization of the animal model.

- Figure 1f is not mentioned in chronological order in text (as are others, too, later)

- Figure 4g/ ext figure 8g: There seems to be no quantification of the 6-months control group shown in Figure 4g. Also, please separate the conditions more clearly, it is impossible to determine which conditions are quantified and shown in extended Figure 8g and to which of the various histologies shown they refer.

- Line 287: No figure number is given for panels h and i.

- A lot of figure panels are not shown in order of appearance - maybe to make a readers' life easier this could be amended? A good example is for instance figure 6b and extended figure 12h – what is the difference between these two cohorts?

Reviewer #2 (Remarks to the Author):

Benvie et al. report that Pdgfr1b expression is induced in inguinal white adipose tissue (iWAT) with aging and that both male and female PDGFRbeta conditional KO mice (Sma) exhibit enhanced beige fat activation and defense of core body temperature in aged but not young mice. Brown fat activation does not appear to be altered in these mice. Pharmacologic blockade of PDGFR1b with imatinib in 6 month old mice is also associated with increased beige fat biogenesis and core body temperature. Treating young mice with PDGF-BB or generating Sma-conditional transgenic mice that express a constitutively active mutant of PDGFR1beta leads to impaired beige fat activation in response to cold temperatures. Adipocyte progenitor cells (APCs) from iWAT did not exhibit age-dependent increases in cellular senescence genes, suggesting that the role of PDGFRbeta on beige fat biogenesis is unlikely to be mediated through senescence. Alternatively, the authors show that anti-IL-33 antibody blockade reverses the induction of beige fat in PDGFR1beta KO mice, and treating the PDGFR1betaD849V mice with recombinant IL-33 induces beige fat (though controls are not shown for this experiment). Aging is associated with increased phosphoSTAT1, and this is reduced in old

PDGFR1betaKO and induced in young D849V mutants. This fits well with prior work from Patrick Seale's lab. Inhibiting STAT1 with fludarabine is associated with increased IL-33 and ILC2s in iWAT as well as increased beige fat in both WT mice. A similar phenotype is seen in PDGFR1betaD849V mice but ILC2s are not reported for this system. Overall, these are elegant and well executed studies, and they provide substantial advances to our understanding of beige fat biology and how PDGF/PDGFR1beta/APCs and IL-33/ILC2 axis coordinate to regulate beige fat function. The rigor in this set of studies is high and bolstered by the use of multiple in vivo and in vitro models. However I do have some significant concerns about the ILC2 gating and analyses:

1. The flow cytometry plots in Extended Data Fig 11 are problematic. The outlier events are hidden from the contour plots, the full gating strategy is not shown (including lineage-negative gates), and the CD25+ CD127+ gates are not convincing for ILC2s. The stain is also lacking IL-33R or some other ILC2 lineage-defining marker, as CD25 and CD127 mark all ILC lineages in adipose. The use of IL13 and IL5 tracks well with what one would expect given the experimental designs and results, which suggests the authors may be correct. However, I would like to see more convincing ILC2 data.
2. FACS plots of ILC2s should be shown in Fig 5 for both the KO and D849V experiments.
3. Numbers of ILC2s per gram of fat should be reported for both the KO and D849V experiments in Fig 5. Reporting percent of CD45+ cells that are ILC2s is an indication of relative abundance but is a function of how many immune cells are present overall. Absolute counts (per gram) are needed to know ILC2 abundance.
4. Fig 5: please show controls treated with vehicle or IL-33
5. Fig 7: what are ILC2 frequencies and numbers in the PDGFR1betaD849V experiment?

Minor: in our experience, we have had the best success with digestion and FACS of adipose tissue with 0.1% Collagenase Type II (Sigma C6885) in DMEM rather than Type I collagenase. They authors might want to consider this for additional work on quantifying ILC2s.

Jonathan R. Brestoff

Reviewer #3 (Remarks to the Author):

PDGFRb signaling and ILC2/IL33 activation both have been shown involved in the regulation of beige adipogenesis based on previous studies. Benvie and colleagues identify the linkage among these factors and have presented interesting findings of how PDGFRb in adult beige APC regulates adipogenesis in both cell-autonomous and non-cell-autonomous manners at different ages. These data may be informative for future studies involving age-dependent regulation of beige APC.

- 1) I have one concern about the time point chosen for aged beige adipogenesis in this study. In Fig1, authors have assessed the function of PDGFRb in beige APC at 6m or 12m of age. iWAT from 6m-old mice has increased PDGFRb expression, and a further upregulated mRNA level was detected in 12m-old mice. Follow-up analysis showed KO of PDGFRb can promote beige adipocyte formation at both 6m and 12m at a different level. However, most of the other analysis authors have performed for APC senescence study was performed using 6m-old-mice, which may not be considered as old by many researchers in the field. It's not clear to me why the authors chose the 6-month model and whether that's because APC at 6m may have senescence/transcriptional identity change ongoing? Or if it's because 6m-old mice have already shown similarities to the 12-m old mice? Authors may consider repeating some of their experiments at 12month-old (imatinib/IL33 Ab treatment/PDGFRb KO model) if mice are still available. Otherwise, authors need to change their conclusion accordingly or indicate a clear rationale for the experiment design.
- 2) Authors in this paper need to provide evidence of PDGFRb activation in all analyses where they applied PDGFRb GOF approaches, which includes the PDGF-BB treatment and PDGFRD849V models. Phosphorylation of PDGFRb should be accessed by western blot on sorted SMA-lineage cells or unsorted stromal vascular fraction from adipose tissue.
- 3) In figure 3 and supplementary Fig 7, authors used the PDGFRbD849V SMA-CreER;R26-mTmG model. I would guess R26-mTmG was built in to check whether PDGFRb can block beige

adipogenesis in cell-autonomous manner. However, they didn't show any lineage reporter stain until Fig 4i. And authors didn't make it clear here whether Fig4i was from the same mice as in Fig3e-k? If so, authors need to provide a better picture indicating that cold-induced UCP1+ beige adipocytes were labeled with membrane GFP from R26-mTmG, and that activation of PDGFRb failed to do so. Current picture doesn't show GFP localized to adipocyte membrane in the control sample. In addition, weak GFP signal can be detected on white adipocyte from PDGFRbD849V mutant, which is inconsistent with previous study demonstrating that PDGFRb activation inhibit white adipogenesis. (Olson, Lorin E., and Philippe Soriano. "PDGFR β signaling regulates mural cell plasticity and inhibits fat development." *Developmental cell* 20.6 (2011): 815-826.)(He, Chaoyong, et al. "STAT1 modulates tissue wasting or overgrowth downstream from PDGFR β ." *Genes & development* 31.16 (2017): 1666-1678.)

4) The Gupta group recently reported that cold challenge transcriptomic changes induction of Il33 expression in DPP4+ PDGFR β + APC. Authors should cite this paper and address how the current findings are consistent or inconsistent accordingly.(Shan, Bo, et al. "Cold-responsive adipocyte progenitors couple adrenergic signaling to immune cell activation to promote beige adipocyte accrual." *Genes & Development*35.19-20 (2021): 1333-1338.)

Several minor issues:

1) For consistency of body temperature analysis: I have noticed that in Fig1f, cold environment exposure decreases the body temperature in 6m control mice from 37.5C to 36C, and from 37.5C to 36.5C in 12m control mice. However, the temperature dropped from 37C to 32C in 6m control mice in the authors' previous paper (Fig. 1c, Berry, Daniel C., et al. "Cellular aging contributes to the failure of cold-induced beige adipocyte formation in old mice and humans." *Cell metabolism* 25.1 (2017): 166-181). Can the authors explain this inconsistency?

2) For some immunofluorescence stains in this manuscript especially those applied to lineage tracing analysis, authors need to provide representative pics in higher magnification. It's difficult to discern co-expression based on current images.

3) For tamoxifen treatment, in current study authors administer 50mg/kg TMX for 2 days for PDGFRb LOF or GOF study. Is it known that this dosage is high enough to induce Cre/PDGFRb GOF or LOF in all SMA-APC?

4) Authors may consider decreasing the dot size in many of their bar graphs to show individual dots clearly.

5) Typo at Line 70: Sma(Acta2)

REVIEWER COMMENTS

Reviewer #1 (Remarks to the Author):

In this manuscript, the authors investigate the effect of elevated age, at 6 and 12 months, on browning capacity in white adipose tissue in mice. They argue that loss of browning capacity may contribute to impaired maintenance of cold tolerance and metabolic health and identify a potential role for PDGFRB signaling and show that this effect is potentially mediated by ILC2 upon release of IL33. Overall, this is a conceptually intriguing analysis that, if verified would contribute a novel and important mechanism to help explain loss of brown adipose thermogenesis during aging. The wide array of datasets is a clear strength, and the authors present a generally promising combination of mouse models to provide some support to their hypothesis which is worth pursuing. However, I have a number of reservations regarding this study that in my view would need to address rather extensively to make this a fully convincing analysis. Specifically, I believe some conclusions are not entirely supported by data and some of the data representations are in view not ideal to allow readers to appreciate the data and their implications fully. Additionally, the paper could benefit from some careful editing and checking, several extended figures are mixed up, descriptions are sometimes slightly cursory and do not allow full appreciation of the data. Here are a few items that the authors could consider:

- We thank the reviewer for their comprehensive critique and helpful considerations. We believe addressing these concerns significantly improved our study and strengthened our conclusions.
- We agree with the reviewer's concerns regarding whole-body metabolism and the misalignment with some beige fat physiological surrogates between the genetic models and the pharmacological studies. These inconsistencies may be related but not limited to sex and sex steroid bioavailability, systemic administration of various pharmacological agents and recombinant proteins, and the activation and deletion of genes within the smooth muscle compartment within WAT and non-adipose tissues. Furthermore, we believe that it would be cost prohibitive to perform metabolic analysis on all animal models and pharmacological approaches—used within this study—to interrogate energy expenditure, especially under ageing considerations. We also agree with the reviewer that it may be difficult to disentangle brown fat activity from beige fat on energy dynamics, and overall contribution to energy balance. Therefore, throughout the manuscript, where possible, we added statements and discussion points that reflect the utility of the genetic and pharmacological approaches used. We do believe, however, at this point, our advanced mouse modeling and the use of various approaches and methods allow us to arrive at a strong conclusion. Finally, we do think that age associated changes in metabolic responses and its link to beige fat biogenesis is an area of interest that needs further investigation but is perhaps beyond the scope of this manuscript.

- In general terms, only beiging is assessed in the study, but the bulk of thermogenesis (in mice) occurs in the regular brown fat depots, dwarfing the contribution of beige adipocytes by several orders of magnitude. The analysis of the main BAT depots, such as interscapular BAT is rather cursory, what happens to the energy metabolism contribution and thermogenesis in these depots?

- We agree with the reviewer that BAT contributes to energy expenditure; however, we are unaware of approaches that would accurately delineate energy requirements and contributions between beige and BAT. Therefore, we have elected to remove most of the energy metabolism aspects and focus on beige adipocyte development. In some models, we measured BAT weight, morphology and thermogenic gene expression (See Extended Data Figures 3m, 4d,j, and 9d). Further, our previous work, along with others, show that Sma marked smooth muscles cells do not contribute to brown adipocyte formation. Further investigation into how vascular smooth muscle cells contribute to brown fat homeostasis and thermogenesis are critical questions. Also, IL-33 administration does not appear to impact brown fat thermogenesis or ILC2 recruitment within brown adipose tissue (Groups: Chawla and Artis).

- Data on cell fate mapping: While these data are certainly of interest, I believe the phenotypical data are more so. The manuscript is already rather the complex and the authors could consider whether these analyses are an essential part of the manuscript. In my mind, a more thorough analysis of the different mouse models metabolic phenotypes would be rather more critical. This is true for two reasons: 1, given that most publications agree that, while browning of WAT is certainly a response to prolonged cold exposure, its contribution to overall energy expenditure is compared to true BAT depots is rather very minor in mice; and 2: most of the metabolic data are not entirely consistent and outcomes vary between the different mouse models. For example, while fludarabine quite markedly improves cold tolerance, no effect on metabolic health is overserved.

- We have taken the reviewers advice to reduce aspects of the fate mapping data as this concern was shared with reviewer three. We agree with the reviewer's rationale around beige vs BAT on thermogenic contributions and energy balance. While we do observe beige fat appearance in response to genetic and pharmacological modeling, we believe that metabolic data may be more complex than initially observed or appreciated. Because Sma cells are expressed throughout the body and pharmacological agents are provided systemically, metabolic measurements could be exaggerated or masked by these effects; thus, not providing a clear resolution or readout for thermogenic fat activation. Because of this key challenge, we have elected to focus on beige fat appearance in twelve-month-old mice in response to modulating *Pdgfrb* gene expression and its signaling pathway.

- Line 85, and using relatively young animals as a model of aging throughout the manuscript: Conceptionally, examining animals at 6 months and in some cases up to an age of 12 months is not really aging research in mice and one can hardly claim rejuvenation of 6-months old mice that are still fully fertile and under normal circumstances even completely healthy. While I agree that the notion of what age defines aging may be somewhat debatable, onset of senescence certainly does not occur at such early ages and I believe the analysis do not convincingly allow for such conclusions. Experimentally, at the very least a time course analysis of animals aged up to 24 months, which is often a standard for aging research

on mice, should be conducted to determine what happens to browning capacity in the post-fertile period of the animal's life. Specifically, expression of *Pdgfrb* should be examined at later stages of life to be correlated with functional markers of browning (ideally also in humans, as 49.33 years is hardly considered very old in people). If such analyses are not possible I very recommend to remove statements that the concept of aging / rejuvenation is addressed from the manuscript.

- We agree with the reviewer that it could be considered debatable when the onset of aging occurs. Further, we are not trying to imply that ageing is defined by a single point(age) or that six-month-old mice are old, we apologize if this was the impression we developed. Rather, we consider aging as a natural process occurring throughout the lifespan of the organism. In this scenario, biological processes within tissues or cells can display aging and cellular senescent-like characteristics throughout this process, thereby impacting specific biological responses, tissue function, and overall homeostasis. Using defined markers of cellular senescence such as SA- β -gal and p16^{ink4a} we previously identified and confirmed herein—using RNA-seq—that adipose tissue SVF can display a senescent-like aging phenotype. We and others have shown that beige fat fails to develop between 6 and 12 months of age (Patrick Seale: Prdm16), Daniel Berry: Cellular Aging, Nichole Rogers: Aging beige, Shingo Kajimura: Mitochondria Lipoylation Beige Fat) depending on thermogenic stimulus. Moreover, Saito's group demonstrated that beige fat biogenesis in humans declines steadily with age, with dampened abilities occurring in the mid-30's (Masayuki Saito: Human Beige Fat). Coinciding with beige fat failure is the highest pervasiveness of obesity and the initiation of various metabolic pathologies including type 2 diabetes (Masayuki Saito: Obesity). Thus, we believe it is critical to understand how beige fat fails in these early settings to identify mechanisms to restore thermogenic function, as soon and as much as possible, to help prevent or revert WAT expansion.

To understand the ageing phenomenon more closely and not overstate, we have done the following:

1. New figure 1: To demonstrate that beige adipocyte development fails as early as 6 months, we performed a cold-exposure time course showing beige adipogenic failing as early as 6 months and remains undetectable in aged animals (24 months). Also, we performed bulk RNA sequencing on SVF from 2- and 12-month-old mice. The sequencing data revealed that over 3,400 genes changing in response to aging. Moreover, these changes reflect genes in p38/Mapk and p53 pathways, which we have showed to contribute to beige APC adipogenic decline (Cellular Aging).
2. We have restricted the use of the word “old, age or ageing” and denoted exact age of the animal (mouse and human) within the study or used the word juvenile and adult (but limited use as well). We also removed several of the senescence tests to focus more on the role of *Pdgfrb* within APCs reduce the emphasize of age-related phenotyping.
3. We have repeated and reconstructed all ageing experiments to be performed at 1 year (12-months) of age. These data mirror the 6-months data collected in the previous version.

- Figure 1: I am not sure I understand the contradicting observations described in the figure: While cold exposure reduces PDGF-BB levels, aging does too. Both would suggest reduced signaling activity through this molecule in both conditions (cold, aging). However, the deletion of the receptor enhances beigeing/metabolic activity, which does not fit the changes in the level of the ligand.

- We apologize for the confusion; this dataset has been removed. We will evaluate Pdgf-BB regulation in future studies.

- Figure 1 and extended figures 3 and 4: The way the data are presented for the 6-months group is rather difficult to put into context with the 2-months cohort (extended Figure 4) and the 12-months group (extended Figure 3) as data are presented in various styles, colors and combinations. In order to avoid misleading interpretations, the authors need to present the data in a consistent way, meaning that temperatures, body weights, adiposity calculations, and glucose levels, etc., should all be reported in the same way and style, with the same axis and statistical comparisons. Related to this, the authors state in lines 99-101 that all metabolic parameters are improved in mutant mice, which is not actually what is shown: Animals of both genotypes show the expected responses to cold. The correct statistical comparison here should be between the genotypes, and here no differences are observed, meaning that there is no effect of the genotype. In summary, I find the presentation of the data somewhat misleading as this comparison is not actually made (or shown) in any of the panels 1g-1i. Moreover, for all age groups, the authors sometimes show the non-cold-exposed control groups, but not for all datasets. Why is this? Temperature data seem to be available for all three ages including the respective control groups, so I think the rest of the data should also be shown. Related to this: Quantifications of browning of multilocular iWAT or Ucp1+ iWAT or lipid droplet size in the different age groups and genotypes seems to be missing. The study hinges on differences in browning, so I think a formal quantification of the histological analyses that are shown to support this statement is warranted.

- We apologize for the confusion, and it was not our intention to be misleading and thank you for your helpful remedies and suggestions. As recommended, all graphs have been adjusted to contain similar axes, include RT datapoints (when applicable), and multiple statistical comparisons. We added beige adipocyte area quantification (Ucp-1 fluorescence quantification) to all appropriate datasets. Extended data fig. 3 contains beige fat physiological surrogates from 2-, 6-, and 12-month-old mice within the same figure for ease of comparison.

- Extended Figure 2: Data of a time course of body temperature in female mice is shown, but only one time point (final) in males in the main figure 1. What about the other time points in the male mice of the main figure 1? Also, the blood glucose levels in females seem the same in both genotypes after cold, while beigeing markers are increased. Wouldn't this argue somewhat against a role for Pdgfrb in metabolic health in female mice? The authors should assess circulating lipids in both sexes as also the fat depots are smaller, which would point towards increased lipid mobilization. There are several possible ways of addressing this, for instance: A minimum would in my view include analysis of HSL phosphorylation and ATGL levels in WAT, measurement of free fatty acids and triglycerides in circulation, and better still would be indirect calorimetry of the animals to determine whether certain nutrients are preferentially metabolized in the KO mice in cold. This would be important to confirm that indeed increased metabolic turnover occurs as a result of increased browning of WAT.

- Thank you for the helpful comments and we have removed metabolic comparisons between male and female mice for reasons denoted above. We appreciate the points and suggestions on lipolysis and circulating fatty acids but to further investigate these notions here, we believe is beyond the scope of the current findings. Thus, we removed this type of data from the current manuscript to maintain focus and clarity. We hope to pursue differences between male and female being and metabolic responses in the future.

- Figure 1j: What happens to expression of *Pdgfrb* at this later time point. In theory, if beige adipocytes are formed better in the KO model, wouldn't one expect to see more adipocytes with deletion of the gene due to the induction. During the six-months phase after induction one would expect are certain level of turnover of adipocytes.

- We found that *Pdgfrb* is continually elevated in aging *Sma+* beige APCs (See figure 1c and extended data figures 1d,k and 2a). The ablation of *Pdgfrb*, at specific ages, show the propensity of more cold-induced beige adipocytes emanating from non-*Sma+* APCs; thus, an alternative population of cells is required. Our initial evidence suggested adipocytes and further studies are required to verify. Yet, it is unclear if there is more adipocyte turnover as the reviewer suggested. This would be an interesting concept to revisit in future studies examining beige adipocyte turnover vs perdurance and frequency of cold temperature exposure.

- Extended Data Fig. 4, lines 122-133: The temperature data in the young mice are very confusing. It looks like young, two-months-old, mice of both genotypes are unable to defend their body temperature during a cold challenge, as body temperature decreases in both genotypes after cold (Extended Figure 4b). The same seems to be true for 6-months old mice. Then suddenly mutant mice aged 12 months (main Figure 1f) gain the ability to defend their body temperature? These data in my view are difficult to reconcile with the concept that that authors propose in their study. To make a convincing argument here, the body temperature data of all age groups and all genotypes with and without cold exposure need to be compared among each other. It looks like the drop in young mice might be less pronounced but this would require a very careful statistical analysis of all data and housing conditions, time of day of measurements, etc. This would need to be documented very thoroughly in order to be able to determine whether such comparisons, if they were not done all at the same time, are indeed permissible.

- We have corrected this issue by showing RT data (see extended data figures 2b-e), rescaling-matching axes, and showing rectal temperatures throughout the cold exposure timeframe. For consistency purposes, we have focused more on beige fat development rather than beige fat physiological surrogates, which may not be indicative of beige fat appearance or performance. Towards this end, we only evaluated the *Pdgfrb*-KO mouse model across 2-, 6- and twelve-months of age (see figure 2). In figure 3, please note changes in rectal temperatures from RT to cold exposure is $\sim 1/2$ degree whereas at 6 and 12-months a larger reduction is observed (1.5-2degrees)

- Figure 2/ extended figure 5: I am a bit doubtful about the specificity of imatinib. While the authors cite a 1996-paper on the ability of the substance class to inhibit PDGF receptors, I believe more recent studies have established that it blocks a number of enzymes with tyrosine kinase activity. At the very least, an analysis of potential target signaling pathways is needed. This includes known targets, i.e. Pdgfrb signaling (this study), bcr-abl signaling (for instance PMID 21098337) and c-Kit (PMID 17607922). As a side note, authors should re-evaluate their statements: They mention that only blood glucose is also reduced in the young cohort when in fact, the only white fat depots to be reduced in size (IGW) is equally affected at both ages, overall somewhat contradicting the statement in lines 154-155.

- We agree with the reviewer's concerns surrounding the specificity of imatinib. To compare and validate imatinib alignment with Pdgfrb genetics, we chose a more selective and potent Pdgfrb inhibitor, SU16f. As recommended, in extended data fig. 9, we show that both imatinib and SU16f effectively downregulate Pdgfrb genes within WAT depots. We have evaluated beige fat development in twelve-month-old mice using imatinib and SU16f. We observe strong parallels in beige fat appearance and gene expression. Again, we strayed away from examining metabolic beige fat surrogates due to the systemic administration of imatinib and SU16f. We further evaluated the overlap between imatinib and SU16f by assessing Stat1-phosphorylation, ILC2 accrual, and ILC2 activation. We found that both imatinib and SU16f reduced Stat1-phosphorylation within twelve-month-old Sma+ beige APCs. Moreover, ILC2 accrual and ILC2 activation also appeared to be augmented in response to imatinib and SU16f treatments. Similar to our genetic Pdgfrb-KO studies, we find that imatinib and SU16f are not additive or synergistic in beige fat development and appear comparable to vehicle treated mice. Overall, the data suggest that imatinib and SU16f functionally mimic the aspects of genetic Pdgfrb deletion in twelve-month old mice (see figure 7).

- Figure 3: Similar to my previous comment, the authors need to show PDGF-BB and overexpression of the constitutively active receptor act on iWAT and alter PDGF BB signaling to establish a direct effect within the tissue.

- We have performed immunoblotting on Pdgfrb expression, phosphorylation of Pdgfrb and stat1. We find that Pdgfrb expression increases with age. Moreover, we find that Pdgfrb is phosphorylated in response to Pdgf-bb treatment. The constitutively active model also showed elevated basal phosphorylation of Pdgfrb (see extended data figures 1d, 4a,4g, and 8a).

- Figure 5: What are IL33 levels in BAT and iWAT of 6- and 12-months old mice compared to younger mice and compared to circulating levels (if any circulates)? In order to link changes of Pdgfrb to IL33 it would in my view be important to investigate the entire signaling cascade. It would also help strengthen the notion that an aging-like phenotype may already occur in relatively young mice. I am surprised by the observation that mice with constitutively active PDGFRB show no differences in ILC2s when this same model features a very pronounced cold intolerance as shown in figure 3. How do the authors explain the discrepancy?

- Thank you for this comment. As observed in the newly added RNA-sequencing data, we did not observe changes in IL-33 expression within iWAT depots between

2- and 12-month-old mice. Therefore, we investigated cold-temperature induced changes in IL-33. Indeed, we observed that IL-33 is upregulated by cold-exposure which is extinguished in 12-month-old mice compared to 2-month-old mice. We found that *Pdgfrb* deletion releases this blockade on IL-33, allowing IL-33 to be expressed in response to cold exposure. Correspondingly, we observed changes in ILC2 numbers reflecting changes in IL-33 availability. We repeated the experiments in constitutively active *Pdgfrb* mouse model using the ILC2 isolation method described by reviewer 2. This methodology improved our collection of ILC2 accrual and reveal distinct changes in the ILC2 population within the various models and settings (see figures 4b and c and extended data figure 6a-d). Of note, these populations were further evaluated for ILC2 markers *Gata3* and *IL33R* (see extended data figure 6e).

- Figure 6: I am unsure about the analysis of Stat1 phosphorylation: A thorough analysis would require assessment of basal / non-phosphorylated as well as phosphorylated forms of the Stat1 protein, for instance by western blot or staining of sorted cells on a cover slip. This links to a previous comment: The authors would need to better establish altered PDGF signaling in their models. As they have also used isolated SVF from the animal models I believe it should be feasible to more clearly assess this in an in vitro cell culture system with inducible deletion/ overexpression or recombinant PDGF-BB. Besides I can't seem to find a methods description of the flow cytometry analysis nor any of the controls used to verify antibody specificity. Intracellular stainings are rather challenging.

- Thank you! We have performed immunoblotting to show that Stat1 is phosphorylated by *Pdgfrb* in the context of adipose tissue (see extended data figure 8a). We have updated the methods to describe the flow cytometric analysis including controls more accurately.

Minor: These issues have been removed or corrected, thank you!

- Line 70: Typo, I think it is *Acta2*, not *Acat2*

- Figure 1i: What is “% delta WAT”? It is only defined as “adiposity” in the figure legend. Please provide more details on how this was calculated and what measurements or data were used for this. Are these data based on body composition analyses by NMR/DEXA? If so, the two main fractions, i.e. fat mass, lean/muscle mass) should be reported. Overall, this type of body composition analysis would be a key element of a convincing metabolic characterization of the animal model.

- Figure 1f is not mentioned in chronological order in text (as are others, too, later)

- Figure 4g/ ext figure 8g: There seems to be no quantification of the 6-months control group shown in Figure 4g. Also, please separate the conditions more clearly, it is impossible to determine which conditions are quantified and shown in extended Figure 8g and to which of the various histologies shown they refer.

- Line 287: No figure number is given for panels h and i.

- A lot of figure panels are not shown in order of appearance - maybe to make a readers' life easier this could be amended? A good example is for instance figure 6b and extended figure 12h – what is the difference between these two cohorts?

Reviewer #2 (Remarks to the Author):

Benvie et al. report that *Pdgfr1b* expression is induced in inguinal white adipose tissue (iWAT) with aging and that both male and female *PDGFRbeta* conditional KO mice (Sma) exhibit enhanced beige fat activation and defense of core body temperature in aged but not young mice. Brown fat activation does not appear to be altered in these mice. Pharmacologic blockade of *PDGFR1b* with imatinib in 6 month old mice is also associated with increased beige fat biogenesis and core body temperature. Treating young mice with PDGF-BB or generating Sma-conditional transgenic mice that express a constitutively active mutant of *PDGFR1beta* leads to impaired beige fat activation in response to cold temperatures. Adipocyte progenitor cells (APCs) from iWAT did not exhibit age-dependent increases in cellular senescence genes, suggesting that the role of *PDGFRbeta* on beige fat biogenesis is unlikely to be mediated through senescence. Alternatively, the authors show that anti-IL-33 antibody blockade reverses the induction of beige fat in *PDGFR1beta* KO mice, and treating the *PDGFR1beta*D849V mice with recombinant IL-33 induces beige fat (though controls are not shown for this experiment). Aging is associated with increased phosphoSTAT1, and this is reduced in old *PDGFR1beta*KO and induced in young D849V mutants. This fits well with prior work from Patrick Seale's lab. Inhibiting STAT1 with fludarabine is associated with increased IL-33 and ILC2s in iWAT as well as increased beige fat in both WT mice. A similar phenotype is seen in *PDGFR1beta*D849V mice but ILC2s are not reported for this system. Overall, these are elegant and well executed studies, and they provide substantial advances to our understanding of beige fat biology and how PDGF/*PDGFR1beta*/APCs and IL-33/ILC2 axis coordinate to regulate beige fat function. The rigor in this set of studies is high and bolstered by the use of multiple in vivo and in vitro models. However I do have some significant concerns about the ILC2 gating and analyses:

- We thank the reviewer for their helpful comments and strategies. We believe implementing these strategies strengthen the ILC2 findings and their linkage to beige adipogenic failure during ageing.

1. The flow cytometry plots in Extended Data Fig 11 are problematic. The outlier events are hidden from the contour plots, the full gating strategy is not shown (including lineage-negative gates), and the CD25+ CD127+ gates are not convincing for ILC2s. The stain is also lacking IL-33R or some other ILC2 lineage-defining marker, as CD25 and CD127 mark all ILC lineages in adipose. The use of Il13 and IL5 tracks well with what one would expect given the experimental designs and results, which suggests the authors may be correct. However, I would like to see more convincing ILC2 data.

We have provided full gating strategies including outliers and updated methodologies. Of note, we did not perform lineage negative sorting prior to Cd25 and Cd127 evaluation, rather from bulk Cd45+ cells. We have provided staining of Gata3 and IL-33R identifying these cells as ILC2s (see extended data figure 6).

2. FACS plots of ILC2s should be shown in Fig 5 for both the KO and D849V experiments.

We have added FACS plots for all studies (see extended data figures 6c, 6d, 6h, 7b, 7g, 8c, 8h, and 9k) .

3. Numbers of ILC2s per gram of fat should be reported for both the KO and D849V experiments in Fig 5. Reporting percent of CD45+ cells that are ILC2s is an indication of relative abundance but is a

function of how many immune cells are present overall. Absolute counts (per gram) are needed to know ILC2 abundance.

We now show both ILC2 frequency and abundance.

4. Fig 5: please show controls treated with vehicle or IL-33

We have added these datasets (see figure 5g).

5. Fig 7: what are ILC2 frequencies and numbers in the PDGFR1betaD849V experiment?

We have added these datasets (see extended data figure 6g and 6h).

Minor: in our experience, we have had the best success with digestion and FACS of adipose tissue with 0.1% Collagenase Type II (Sigma C6885) in DMEM rather than Type I collagenase. They authors might want to consider this for additional work on quantifying ILC2s.

THANK YOU! Using this new method helped significantly increase our yield of ILC2s.

Jonathan R. Brestoff

Reviewer #3 (Remarks to the Author):

PDGFRb signaling and ILC2/IL33 activation both have been shown involved in the regulation of beige adipogenesis based on previous studies. Benvie and colleagues identify the linkage among these factors and have presented interesting findings of how PDGFRb in adult beige APC regulates adipogenesis in both cell-autonomous and non-cell-autonomous manners at different ages. These data may be informative for future studies involving age-dependent regulation of beige APC.

- We thank the reviewer for their helpful and constructive critiques. We believe these adjustments significantly improved our findings and support our overall hypothesis.

1) I have one concern about the time point chosen for aged beige adipogenesis in this study. In Fig1, authors have assessed the function of PDGFRb in beige APC at 6m or 12m of age. iWAT from 6m-old mice has increased PDGFRb expression, and a further upregulated mRNA level was detected in 12m-old mice. Follow-up analysis showed KO of PDGFRb can promote beige adipocyte formation at both 6m and 12m at a different level. However, most of the other analysis authors have performed for APC senescence study was performed using 6m-old-mice, which may not be considered as old by many researchers in the field. It's not clear to me why the authors chose the 6-month model and whether that's because APC at 6m may have senescence/transcriptional identity change ongoing? Or if it's because 6m-old mice have already shown similarities to the 12-m old mice? Authors may consider repeating some of their experiments at 12month-old (imatinib/IL33 Ab treatment/PDGFRb KO model) if mice are still

available. Otherwise, authors need to change their conclusion accordingly or indicate a clear rationale for the experiment design.

- We thank the reviewer for their helpful comments and apologize for the confusion behind the rationale to use 6-month-old studies. Briefly, in 2017, we identified that Sma+ cells, which can serve as white and beige adipocyte progenitors (APCs), acquire a cellular senescent-like phenotype which led to the impaired beige fat development with age. Interestingly, this phenomenon occurred as early as 6-months of age in mice which corresponded to human data, indicating that human beige fat development begins to decline by the mid-30's. We also validated that beige fat failure was diminished in 1 year old mice. Moreover, we identified that p38 phosphorylation was a major regulator of beige adipogenic failure and potentially cellular aging. However, we were unable to identify the upstream signaling mechanism controlling p38 phosphorylation in APCs to control the cellular aging like phenotype. Thus, when we identified that Pdgfrb was upregulated in aging animals, we hypothesized that Pdgfrb might mediate p38 phosphorylation. Therefore, we begin investigating six-month-old mice to understand/elucidate early signals in beige adipogenic failure. However, based on our experimental profiling and data collection, Pdgfrb does not appear to mediate cellular senescence.
- In consideration with yours and reviewer 1's comment, we have repeated all experiments on 12-month-old mice. These data recapitulated our six-month studies, and we believe these new studies strengthen the aging concept. Yet, we have dampened the word use of "old, age, and aged" and used denoted times of age or ageing or juvenile and adult.

2) Authors in this paper need to provide evidence of PDGFRb activation in all analyses where they applied PDGFRb GOF approaches, which includes the PDGF-BB treatment and PDGFRD849V models. Phosphorylation of PDGFRb should be accessed by western blot on sorted SMA-lineage cells or unsorted stromal vascular fraction from adipose tissue.

We have added immunoblots showing:

1. Extended data Figure 1: Pdgfrb expression between 2 and 12-months of age.
2. Extended data figure 4: Pdgfrb phosphorylation in SVF of Pdgf-bb treated and PdgfrbD849V mice.
3. Extended Data Figure 8: Stat1 phosphorylation in response to Pdgfrb constitutive activation.

3) In figure 3 and supplementary Fig 7, authors used the PDGFRbD849V SMA-CreER;R26-mTmG model. I would guess R26-mTmG was built in to check whether PDGFRb can block beige adipogenesis in cell-autonomous manner. However, they didn't show any lineage reporter stain until Fig 4i. And authors didn't make it clear here whether Fig4i was from the same mice as in Fig3e-k? If so, authors need to provide a better picture indicating that cold-induced UCP1+ beige adipocytes were labeled with membrane GFP from R26-mTmG, and that activation of PDGFRb failed to do so. Current picture doesn't show GFP localized to adipocyte membrane in the control sample. In addition, weak GFP signal can be detected on white adipocyte from PDGFRbD849V mutant, which

is inconsistent with previous study demonstrating that PDGFRb activation inhibit white adipogenesis. (Olson, Lorin E., and Philippe Soriano. "PDGFR β signaling regulates mural cell plasticity and inhibits fat development." *Developmental cell* 20.6 (2011): 815-826.)(He, Chaoyong, et al. "STAT1 modulates tissue wasting or overgrowth downstream from PDGFR β ." *Genes & development* 31.16 (2017): 1666-1678.)

- We apologize for the confusion. Indeed, these tissue samples originated from the animals generated from the original manuscript Fig. 3. Using the RFP lineage tracing mouse model system, we have provided higher magnification of fate mapping analysis (see extended data figure 5g). As suggested by reviewer 1, we have limited our lineage tracing studies to just the initial observation that *Pdgfrb* deletion facilitates beige fat biogenesis independent of identified *Sma*⁺ beige APCs. We agree with reviewer 3 that *Pdgfrb* regulates adipose tissue development and homeostasis and more studies directed at understanding how *Pdgfrb* regulates APC lineage decisions will be insightful.

4) The Gupta group recently reported that cold challenge transcriptomic changes induction of *Il33* expression in *DPP4*⁺ *PDGFR β* ⁺ APC. Authors should cite this paper and address how the current findings are consistent or inconsistent accordingly.(Shan, Bo, et al. "Cold-responsive adipocyte progenitors couple adrenergic signaling to immune cell activation to promote beige adipocyte accrual." *Genes & Development* 35.19-20 (2021): 1333-1338.)

Thank you for noting this study, we have added a discussion point around this new study.

Several minor issues:

1) For consistency of body temperature analysis: I have noticed that in Fig1f, cold environment exposure decreases the body temperature in 6m control mice from 37.5C to 36C, and from 37.5C to 36.5C in 12m control mice. However, the temperature dropped from 37C to 32C in 6m control mice in the authors' previous paper (Fig. 1c, Berry, Daniel C., et al. "Cellular aging contributes to the failure of cold-induced beige adipocyte formation in old mice and humans." *Cell metabolism* 25.1 (2017): 166-181). Can the authors explain this inconsistency?

- While performing these studies, we noticed changes in temperature defense as well. Mice were relocated from UTSW to Cornell and due to pathogens, these mice were rederived. There are also differences in vivarium housing conditions such as light-dark cycle, basal room temperature, and the use of ventilated cages all of which could be involved in changing metabolic responses including beige fat biogenesis.

2) For some immunofluorescence stains in this manuscript especially those applied to lineage tracing analysis, authors need to provide representative pics in higher magnification. It's difficult to discern co-expression based on current images.

- We have added high magnification of fate mapping analysis and further quantification of fate mapping and beige fat area (see extended data figure 5g and 5h) .

3) For tamoxifen treatment, in current study authors administer 50mg/kg TMX for 2 days for PDGFRb LOF or GOF study. Is it known that this dosage is high enough to induce Cre/PDGFRb GOF or LOF in all SMA-APC?

- Yes, we have added these datasets and referenced our previous studies demonstrating recombination efficiency between reporter and endogenous Sma expression (see extended data figure 1i and 1j).

4) Authors may consider decreasing the dot size in many of their bar graphs to show individual dots clearly. Corrected, **thank you!**

5) Typo at Line 70: Sma(Acta2) Corrected, **thank you!**

REVIEWER COMMENTS

Reviewer #1 (Remarks to the Author):

The authors have significantly improved their manuscript and have addressed most of my comments, some of which by removing the dataset in question. Instead they provide new datasets, some of which I rate much more robust and convincing than what was shown in the previous version. I would, however, urge the authors to revisit the works from Cannon/Nedergaard regarding absolute quantification of UCP1 protein in WAT and BAT of mice (and the whole animal) and their impact on UCP1-dependent thermogenesis output. While one cannot infer actual thermogenesis from UCP1 protein levels, I believe the marked difference in UCP1 levels between white and brown fat certainly implies a much less effective level of thermogenesis simply due to the lack of the responsible machinery.

While I wholeheartedly agree that a full assessment of energy metabolism in all models presented in the manuscript would be beyond its scope, I would have thought that an attempt to verify the physiological impact of enhanced browning of WAT in the context of aging would be important. A simple measurement of body temperature may have been feasible. But then again this may be a point of discussion and does not strictly impact the quality and high interest of the revised and presented manuscript.

Beyond this, the authors did respond to all my comments very well, and only some minor items could be considered which partially arise from the newly incorporated data:

- New figure 1: What is the expression of Pdgfrb and p16 (and/or other senescence markers) in the samples shown in 1a/1b? i.e. at 24 months of age?
- New figure 1: It would be helpful to highlight the enrichment of senescence/ aging genes a bit more clearly. The heatmap in view is not very informative as it only shows that differentially expressed genes are present. More informative might be a list or graph to indicate which genes are differentially expressed in the p53 and MAPK pathways which the authors mention in the text.
- My comment to previous figure 6: The idea was to assess non-phosphorylated STAT1. While an additional Western blot of phospho-Stat1 is clearly important and nicely confirms the original data, a proper control experiment would include the non-phosphorylated protein. I am not sure whether a good antibody is available, though.

Reviewer #2 (Remarks to the Author):

While this manuscript still has many strengths and is quite interesting overall, unfortunately the gating strategy used to define ILC2s precludes identification of this cell type in all of the studies. The authors gate on CD45+ cells and then plot CD25 against CD127 to identify a population of double-positive cells, which are IL-33R+ and GATA3+. These cells will include both ILC2s and Th2 cells (and possibly other cells too), which are also found in iWAT. ILC2s are defined as lineage-negative cells that lack T cell, B cell, and myeloid lineage markers. The minimum lineage-negative gating strategy required to define ILCs is live CD45+ CD3- CD19- CD11b- CD11c- cells. In WAT and BAT, ILC2s are Lin- cells that are NK1.1- CD25+ CD127+ CD90+ IL-33R+ GATA3+ (not an exhaustive list).

In addition, the FACS plots suggest there are issues with the compensation matrices used.

I would suggest that the authors either omit the FACS data and all associated conclusions about ILC2s, or they should re-run the experiments reporting ILC2s to properly define them.

Reviewer #3 (Remarks to the Author):

I just have one minor concern here about the body temperature change due to the cold challenge. I'm not sure if the mice relocation will make that dramatic difference. Most of my other comments were nicely addressed in the current version. I think it's good to be considered for publication at this time.

REVIEWER COMMENTS

Reviewer #1 (Remarks to the Author):

The authors have significantly improved their manuscript and have addressed most of my comments, some of which by removing the dataset in question. Instead they provide new datasets, some of which I rate much more robust and convincing than what was shown in the previous version. I would, however, urge the authors to revisit the works from Cannon/Nedergaard regarding absolute quantification of UCP1 protein in WAT and BAT of mice (and the whole animal) and their impact on UCP1-dependent thermogenesis output. While one cannot infer actual thermogenesis from UCP1 protein levels, I believe the marked difference in UCP1 levels between white and brown fat certainly implies a much less effective level of thermogenesis simply due to the lack of the responsible machinery.

While I wholeheartedly agree that a full assessment of energy metabolism in all models presented in the manuscript would be beyond its scope, I would have thought that an attempt to verify the physiological impact of enhanced browning of WAT in the context of aging would be important. A simple measurement of body temperature may have been feasible. But then again this may be a point of discussion and does not strictly impact the quality and high interest of the revised and presented manuscript.

- We thank the reviewer for noting the improved data set and for their thoughtful comments that lead to this revision! We appreciate the reviewer's feedback and will address the *Pdgfrb*-induced beige adipocytes thermogenic and metabolic concerns in juvenile and adult animals in a subsequent manuscript.

Beyond this, the authors did respond to all my comments very well, and only some minor items could be considered which partially arise from the newly incorporated data:

- New figure 1: What is the expression of *Pdgfrb* and *p16* (and/or other senescence markers) in the samples shown in 1a/1b? i.e. at 24 months of age?

Please see the qPCR data below: Of note, the expression is from total iWAT from cold exposed mice whereas the data figure in Fig. 1c is on isolated *Sma+* cells from mice maintained at room temperature. We have not added this data to the manuscript.

- New figure 1: It would be helpful to highlight the enrichment of senescence/ aging genes a bit more clearly. The heatmap in view is not very informative as it only shows that differentially expressed genes are present. More informative might be a list or graph to indicate which genes are differentially expressed in the p53 and MAPK pathways which the authors mention in the text.

We have provided two more additional assessments in Extended Data Fig. 1 d, and e. We added a gene ontology analysis for senescence genes and SASP markers along with a complementary heat map. For ease, we have provided references for each marker showing their correspondence and involvement regarding senescence and ageing.

Il10: Interleukin 10 or human cytokine synthesis inhibitory factor has been shown to be a SASP factor, specifically during replicative senescence.

<https://doi.org/10.1016/j.cellsig.2019.109445>

<https://doi.org/10.1177/15353702209603>

<https://doi.org/10.1016/j.cell.2017.11.019>

<https://doi.org/10.1038/ncomms8847>

<https://doi.org/10.1038/s42255-020-00305-3>

Il1b: Interleukin-1 beta is a known pro-inflammatory SASP factor that is upregulated during cellular senescence. Il1b binding to its receptor Il-1R has been shown to activate known senescent genes and pathways including NF- κ B and p38 MAPK.

<https://doi.org/10.1038/ncb2811>

<https://doi.org/10.1126/scisignal.3105cm1>

<https://doi.org/10.3389/fneur.2020.00929>

<https://doi.org/10.1159/000504298>

<https://doi.org/10.1016/j.isci.2021.103250>

Sesn2: Sesn2, regulated by p53 and has been shown to decrease with age and deletion of this gene has been shown to cause advanced aging phenotypes.

<https://doi.org/10.1016/j.cmet.2013.08.018>

<https://doi.org/10.1002/emmm.201000097>

<https://doi.org/10.1126/science.1182228>

<https://doi.org/10.1096/fj.201700063R>

Cdk2: Cyclin dependent kinase 2 or Cdk2 is known to regulate cell cycle progression and DNA replication. p21 activation during senescence results in downstream inactivation of CDKs, including CDK2, inhibiting cell cycle progression.

<https://doi.org/10.3389/fcell.2021.645593>

<https://doi.org/10.1038/ncb0110-7>

<https://doi.org/10.1038/ncb2004>

<https://doi.org/10.1158/1541-7786.MCR-14-0163>

Cdkn2a: Cyclin-dependent kinase inhibitor 2A encodes for the cell cycle inhibitor p16 that is often upregulated during aging and a hallmark senescent cell marker.

<http://dx.doi.org/10.1016/j.cell.2017.05.015>

<https://doi.org/10.3389/fcell.2021.645593>

<https://doi.org/10.1093/emboj/cdg417>

<https://doi.org/10.18632/aging.202640>

- My comment to previous figure 6: The idea was to assess non-phosphorylated STAT1. While an additional Western blot of phospho-Stat1 is clearly important and nicely confirms the original data, a proper control experiment would include the non-phosphorylated protein. I am not sure whether a good antibody is available, though.

Our apologies for the confusion: we have added the total Stat1 western blot from 2- and 12-month-old mice (Extended Data Fig. 8a) demonstrating that Stat1 is upregulated. Additionally, our bulk RNA-seq data demonstrated that Stat1 gene expression was upregulated in 12-month-old mice compared to 2-month-old mice. Therefore, we added this dataset to Extended data fig 1d, e as well.

Reviewer #2 (Remarks to the Author):

While this manuscript still has many strengths and is quite interesting overall, unfortunately the gating strategy used to define ILC2s precludes identification of this cell type in all of the studies. The authors gate on CD45+ cells and then plot CD25 against CD127 to identify a population of double-positive cells, which are IL-33R+ and GATA3+. These cells will include both ILC2s and Th2 cells (and possibly other cells too), which are also found in iWAT. ILC2s are defined as lineage-negative cells that lack T cell, B cell, and myeloid lineage markers. The minimum lineage-negative gating strategy required to define ILCs is live CD45+ CD3- CD19- CD11b- CD11c- cells. In WAT and BAT, ILC2s are Lin- cells that are NK1.1- CD25+ CD127+ CD90+ IL-33R+ GATA3+ (not an exhaustive list).

In addition, the FACS plots suggest there are issues with the compensation matrices used.

I would suggest that the authors either omit the FACS data and all associated conclusions about ILC2s, or they should re-run the experiments reporting ILC2s to properly define them.

- We thank the reviewer for noting the improved data set, strengths, and interest. We understand the reviewer's hesitation and restriction around the identity of the immunological cells. Unfortunately, we do not have the animals or resources to redo all flow cytometric analysis on Lin- gating procedures as recommended. Therefore, as proposed by the reviewer, we have removed all flow cytometric analysis of potential ILC2 datasets and have removed all conclusion regarding ILC2 on beige fat development. However, we have kept the mRNA analysis of IL-33, IL-13, and IL-5 as possible indicators of altered type 2 cytokine signaling and immune cell composition. We have adjusted the text and model to reflect these changes and discuss future questions. We believe that these will be important future steps towards understanding beige fat development in ageing mammals. We have also applied compensation matrices to other flow plots such as the Sma and mGFP recombination figure in Extended data Fig. 1k. In agreement with the reviewer, we believe the dataset without the flow cytometric analysis still provides

new and exciting insight into beige adipocyte biology in ageing mice.

Reviewer #3 (Remarks to the Author):

I just have one minor concern here about the body temperature change due to the cold challenge. I'm not sure if the mice relocation will make that dramatic difference. Most of my other comments were nicely addressed in the current version. I think it's good to be considered for publication at this time.

- We thank the reviewer for noting the improved dataset and consideration for publication. We agree with the reviewer and wish we could provide a better explanation for the temperature variation. Of note, the mice were not just relocated but rather *re-derived* to eliminate unacceptable pathogens. Critically, the gut microbiome has been shown to influence thermogenesis; thus, changes in gut microbiota composition and pathogens could provide a possible explanation; however, it may not be the full spectrum.

REVIEWERS' COMMENTS

Reviewer #2 (Remarks to the Author):

The authors have adequately addressed my remaining concerns